

# Flow-induced errors in airborne in-situ measurements of aerosols and clouds

Antonio Spanu[1], Maximilian Dollner[1], Josef Gasteiger[1], T. Paul Bui[2], and Bernadett Weinzierl[1]

[1]University of Vienna (UNIVIE), Aerosol Physics and Environmental Physics, Wien, Austria
[2]NASA Ames Research Center, Mountain View, CA, USA

**Correspondence:** Antonio Spanu (antonio.spanu@univie.ac.at)

**Abstract.** Aerosols and clouds affect atmospheric radiative processes and climate in many complex ways and still pose the largest uncertainty in current estimates of the Earth's changing energy budget.

Airborne in-situ sensors such as the Cloud, Aerosol, and Precipitation Spectrometer (CAPS) or other optical spectrometers and optical array probes provide detailed information about the horizontal and vertical distribution of aerosol and cloud properties.

However, flow distortions occurring at the location where these instruments are mounted on the outside of an aircraft may directly produce artifacts in detected particle number concentration and also cause droplet deformation and/or break-up during the measurement process.

Several studies have investigated flow-induced errors assuming that air is incompressible. However, for fast-flying aircraft, the impact of air compressibility is no longer negligible. In this study, we combine airborne data with numerical simulations to investigate the flow around wing-mounted instruments and the induced errors for different realistic flight conditions. A correction scheme for deriving particle number concentrations from in-situ aerosol and cloud probes is proposed, and a new formula is provided for deriving the droplet volume from images taken by optical array probes, reducing errors by up to one order of magnitude. Shape distortions of liquid droplets can either be caused by errors in the speed with which the images are recorded or by aerodynamic forces acting at the droplet surface caused by changes in the airflow around the instrument. These forces can lead to the dynamic breakup of droplets causing artifacts in particle number concentration and size. Furthermore, an estimation of the critical breakup diameter, as a function of flight conditions is provided.

Experimental data show that flow speed at the instrument location is smaller than the ambient flow speed. Our simulations confirm the observed difference and reveal a size-dependent impact on particle speed and concentration. This leads, on average, to a 25 % overestimation of the number concentration of particles larger than 10 μm diameter and causes distorted images of droplets and ice crystals if the flow values recorded at the instrument are used. With the proposed correction scheme both errors are significantly reduced by a factor 10.

Although the presented correction scheme is derived for the DLR Falcon research aircraft (SALTRACE campaign) and validated for the DLR Falcon (A-LIFE campaign) and the NASA DC-8 (ATom campaign), the general conclusions hold for any fast-flying research airplane.



## 1 Introduction

Aerosol-cloud-radiation interactions are one of the largest uncertainties in current climate predictions (Stocker et al., 2014). The size distribution of cloud and aerosol particles is a crucial parameter for aerosol-radiation and aerosol-cloud interaction (Albrecht, 1989; Rosenfeld and Lensky, 1998; Pruppacher and Klett, 2010). For example, an increase of the fraction of coarse

particles can modify the direct radiative forcing of dust from cooling to warming (Kok et al., 2017).

Airborne in-situ measurements are fundamental to extend our knowledge of cloud and aerosol distributions, especially in the coarse mode. Instruments typically used by the aerosol and cloud community, for measuring coarse particles, are open path or passive-inlet[1] optical particle counters (OPCs) and optical array probes (OAPs). OPCs and OAPs measure particle flux as they record, within a time interval, the number of particles passing through a specific region named sampling area. The flux is

later converted into a concentration using the air flow speed. Therefore, errors in the flow speed are directly affecting the calculated particle and cloud hydrometeor concentrations. For example, a flow speed recorded too low leads to a higher calculated particle concentration. Since the aircraft itself can influence the surrounding air and the flow measurements (Kalogiros and Wang, 2002), airborne measurements are challenging. Flow distortion caused by the fuselage and wings not only impacts the flow velocity but also modifies temperature and pressure from free stream conditions thereby further affecting the aerosol and

cloud measurements. Furthermore, droplets can break up during high-speed sampling resulting in an enhanced concentration of small droplets (Weber et al., 1998). These shattering artifacts may originate from aerodynamic or impaction breakup of cloud droplets and ice particles in and around the aerosol inlet (Korolev and Isaac, 2005; Craig et al., 2013). Large droplets may appear deformed as the result of aerodynamic forces acting on the droplet surface, as studied by Szakall et al. (2009); Vargas and Feo (2010).

Generally, the degree of the artifact depends on the mounting position of the instrument at the aircraft and also on the flight conditions. Effects of a disturbed flow field on observed particle concentrations have been studied, under the incompressible hypothesis (e.g., King, 1984; King et al., 1984; Drummond and MacPherson, 1985; Norment, 1988). However, the assumption that air is incompressible does not hold for measurements on fast-flying aircraft (>100 $\mathrm{ms^{-1}}$). Computational Fluid Dynamics (CFD) models are a powerful tool to study aircraft inlets (e.g., Korolev et al., 2013; Moharreri et al., 2013, 2014; Craig et al.,

2013, 2014)) and sensors (Laucks and Twohy, 1998; Cruette et al., 2000), but are computationally expensive. That is why many studies considered only the instrument itself, but not the combined effect of the aircraft and the instrument.

Recently Weigel et al. (2016) proposed a more general correction method for compressible flow mainly based on thermodynamical calculations. However, this empirical approach is only partially considering the size-dependent effect of particle inertia on the detected concentration. Furthermore, flow disturbances induced by the aircraft wings are not considered by Weigel et al.

(2016).

In this study, the influence of airflow distortion caused by the instrument and aircraft wing is characterized for airborne aerosol and cloud measurements. Under the hypothesis of a compressible flow, this study investigates how different flight conditions

---

[1]Instruments with passive inlets are not actively sampling the air with a pump, instead they rely on the air flow resulting from the wind or the airplane motion.



affect particle concentrations depending on size.

We propose a correction strategy valid for different plane configurations and passive inlet instruments. Moreover, we investigate how water droplets deform when approaching a wing-mounted instrument. Errors affecting the estimation of the droplet volume from OAP images are studied using different approximating formulas. Numerical results are compared with in-situ

measurements collected with a Cloud and Aerosol Spectrometer with Depolarization Detection (CAS-DPOL, Droplet Measurement Techniques (DMT) Inc., Longmont, CO, USA; Baumgardner et al. (2001)) and a second-generation Cloud, Aerosol and Precipitation Spectrometer (CAPS). The analysis is valid for a variety of wing-mounted OPC and OAP instruments used by the aerosol and cloud community. Other potential error sources affecting OPC and OAP measurements like calibration method (Walser et al., 2017), optical misalignment (Lance et al., 2010), or size-dependent sampling area (Hayman et al., 2016) are not

considered in this paper.

The paper is organized as follows: section 2 introduces the methodology. For clarity, we divided the presentation of the results into two parts. The first part (Sec. 3.1) analyzes flow changes around wing-mounted instruments and their effects on derived particle concentrations. Also, a correction strategy is described. The second part (Sec. 3.2) describes a method that allows one to use for OAP measurements a corrected speed of very large particles. It includes an evaluation of a parameterization

of the droplet breakup process, as well as the verification of numerical results with experimental data. Different formulas for calculating the droplet volume and the undisturbed droplet diameter from OAP images are evaluated. The manuscript closes with recommendations (Sec. 4) helping to reduce errors in airborne aerosol and cloud measurements and concluding remarks.

## 2 Methodology

The correction strategy presented in this manuscript is based on numerical simulations of airflow and particle motion and

20 field data collected in 2013 during the Saharan Aerosol Long-range Transport and Aerosol-Cloud-Interaction Experiment (SALTRACE, Weinzierl et al. (2017)). The primary purpose is to quantify flow-induced measurement errors and to present a concentration correction scheme. The proposed correction scheme is later tested with independent datasets collected during two field campaigns, the Absorbing aerosol layers in a changing climate: aging, lifetime and dynamics mission conducted in 2017 (A-LIFE, Weinzierl and ALIFE_Team (2018)), and the Atmospheric Tomography Mission over the years 2016-2018

(ATom-1 to ATom-4, Wofsy et al. (2018)).

### 2.1 Airborne aerosol measurements on-board the DLR Falcon and the NASA DC-8 research aircraft

Our primary analysis focuses on the DLR research aircraft Dassault Falcon 20E (registration D-CMET) and is later applied to the NASA DC-8. Figure 1 shows a sketch of the DLR Falcon with a wing mounted instrument, such as the CAPS. Table 1 gives an overview of the specifications of Falcon and DC-8 including the range of typical aircraft cruise speeds and instruments used

for this study.

The typical altitude range covered by the DLR Falcon is below 12800 m, and aircraft speed ranges from 80 ms$^{-1}$, at low altitude, to 220 ms$^{-1}$ at higher altitude (see Tab. 1). The DLR Falcon is equipped per default with a Rosemount 5-hole





pressure probe model R858A on the tip of the nose-boom (see Fig. 1), referred to as the CMET system in our study. The CMET system measures inflow velocity and direction and has been calibrated using a cone trail (Bogel and Baumann, 1991). Bogel and Baumann (1991) estimate static pressure errors during pilot-induced maneuvers being smaller than 1 %.

The NASA DC-8 can fly at altitudes up to 13800 m with aircraft speeds between 90 and 250 $\mathrm{ms^{-1}}$. During the ATom mission,
the NASA DC-8 was equipped with the Meteorological Measurement System (MMS, Scott et al. (1990)). The MMS hardware consists of three major systems: an air-motion sensing system to measure air velocity with respect to the aircraft, an aircraft-motion sensing system to measure the aircraft velocity with respect to the earth, and a data acquisition system to sample, process, and record the measured quantities (Chan et al., 1998; Scott et al., 1990).

### 2.1.1 Aerosol and cloud instruments

In this section, we describe the instruments used for aerosol and droplet measurements during the different campaigns. For SALTRACE, the DLR Falcon was equipped with a CAS-DPOL mounted under the wing, hereafter named CAS. CAS is a passive inlet OPC (Baumgardner et al., 2001). Other similar open path and passive inlet instruments are the Forward Scattering Spectrometer Probe (FSSP type 100 and 300), the Cloud Droplet Probe (CDP), and the Cloud Particle Spectrometer with Polarization Detection (CPSPD) (Knollenberg, 1976; Lance et al., 2010; Baumgardner et al., 2014). The general measurement
mechanism of an OPC is the following: when a particle passes through the laser beam, it scatters light, which is collected by an optical system and detected by a photo-detector. The resulting signal is then recorded and converted to the instrument-specific scattering cross section. Using scattering theory the particle size can be inverted from this cross section (e.g., Walser et al., 2017).

During A-LIFE and the ATom missions, the CAPS was used as the aerosol and cloud instrument. CAPS is a composed in-
strument consisting of a second generation CAS and a Cloud Imaging Probe (CIP). CIP is an OAP, which was introduced by Knollenberg (1970) and extensively used for droplet and ice-crystal measurements. OAPs measure particle diameters by detecting the shadow formed by the particle passing through a collimated laser beam. The CIP's unit acquires a full particle image by assembling a sequence of image slices. The image acquisition frequency, named shutter speed of the camera, should be proportional to the particle speed. CIP's working mechanism is similar to those other OAPs like 2D-S, HVPS, and SID-2
(Knollenberg, 1981; Lawson et al., 2006; Cotton and Osborne, 2009).

### 2.1.2 Measurement of airflow and flow distortion effects caused by the aircraft fuselage

The DLR Falcon is equipped with four hard points under the wings to carry up to four instruments inside standard canisters (developed by Particle Measuring Systems). Canisters have an outer diameter of $\sim 0.177$ m with a 1.25 m length and are mounted with 3.5-degree angle with respect to the wing (see the green arrow in Fig. 1).
Most wing-mounted instruments are equipped with flow sensors to constrain local conditions. Commonly used sensors are pitot-static tubes, hereafter referred to as pitot tubes (Letko, 1947; Garcy, 1980). Pitot-tubes are usually located to measure flow conditions at the sampling area. A pitot tube measures total pressure $p_{tot}$ and the static pressure $p_s$. $p_{tot}$ is the sum of the static and dynamic pressure $q_c$ and is a measure of total energy per unit volume. Consequently, $p_{tot}$ should not change





around the airplane if dissipative processes, such as a shock wave, do not occur. At small Mach numbers ($M = U/U_{sound}$<0.3, approximately corresponding to $100\,\mathrm{ms}^{-1}$ at sea level), airspeed $U$ can be derived using the incompressible form of Bernoulli's Equation. When the airspeed increases ($M$>0.3), air density cannot be considered independent of velocity. For this reason, a generalized Bernoulli's equation is needed:

$$5 \quad \int_{p_1}^{p_s} \frac{dp}{\rho} + \frac{U^2}{2} = \mathrm{const} \tag{1}$$

$p_1$ is a static reference pressure and the air density $\rho$ is a function of pressure. Using the heat capacity ratio $\gamma$ and $U_{sound} = \sqrt{\gamma p_s/\rho}$ for an adiabatically expanding gas, the following expression can be derived:

$$\frac{p_{tot}}{p_s} = \left(1 + \frac{\gamma - 1}{2} M^2\right)^{\frac{\gamma}{\gamma - 1}} \tag{2}$$

It is assumed that the pitot tube is parallel to the flow. Eq. 2 can be converted to obtain the flow speed

$$10 \quad U = \sqrt{\frac{2\gamma}{\gamma - 1} \frac{p_s}{\rho} \left(\left(\frac{p_{tot}}{p_s}\right)^{\frac{\gamma - 1}{\gamma}} - 1\right)} \tag{3}$$

Therefore, errors of pressure-based airspeed instruments are related to the static pressure as well as to the dynamic pressure (Nacass, 1992). Static pressure errors are typically introduced by disturbances in the flow field around the aircraft and mostly depend on the location and design of the pitot tube (Garcy, 1980). The amount by which the local static pressure at a given point in the flow field differs from the free stream static pressure is called the positional error.

Errors in the dynamic pressure may occur due to flow disturbances caused by the aircraft or by excessive flow angularity, for example when flying with a large angle of attack (>5$^o$). In this last case, which may occur during fast ascent or descents or steep turns of the aircraft, the fluid stream is not anymore parallel to the instrument head and errors occur in both total and static pressure readings (Sun et al., 2007; Masud, 2010). For the DLR Falcon, when flying in normal condition, i.e. avoiding steep turns, the attack angle is small enough to have only a negligible contribution to the error.

The True Air Speed (TAS), i.e. the speed of the air in the free stream, can deviate from the Probe Air Speed (PAS), i.e. the airspeed measured at the probe location. King (1984) estimated the difference between TAS and PAS being smaller than 10 % and varying as the inverse square of the scaled distance from the aircraft nose. However, the estimation of King (1984) relies on an incompressible fluid. Using the incompressible Bernoulli's equation will lead to a 10 % overestimation of airspeed with an 8 % error in pressure as the aircraft approaches transonic speeds ($M \sim 0.8$).

The CAS is equipped with a 17 cm pitot tube, whereas the CAPS has a 24 cm long one. Pressure sensors have been statically calibrated by the manufacturer. Therefore, the positional errors can be estimated using the deviances between the CMET measurements, representing the free stream conditions, and the wing mounted instrument reading. Figure 2 shows a statistical comparison between temperature (a), dynamic pressure (b) and static pressure (c) values recorded by the CMET system at the nose boom (free stream) and by the CAS instrument. In Fig. 2d TAS$_{CMET}$ is compared with the PAS calculated using the pitot tube data according to Eq. 3. Pixels are color coded with the statistical frequency of the binned data. Red lines in Fig. 2a-c are





polynomial fits of the data with calculated R-squares values.

As indicated by the deviation from the 1:1 line (dashed), wing-mounted instruments experience an overpressure on their static sensors (Fig. 2b). Since the total pressure is constant along the plane, a higher static pressure $p_s$ results in lower dynamic pressure $q_c$ (Fig. 2c). Consequently, the calculated PAS is on average 30 % lower than TAS, with a 35 % maximum relative

deviation at higher speed (Fig. 2d). To investigate the discrepancy between PAS and TAS, we use a numerical model.

## 2.2  Numerical models

### 2.2.1  Flow model

As discussed before, the incompressible hypothesis is not valid for fast-flying aircraft ($M > 0.3$) such as the DLR Falcon and the NASA DC-8 and a more general model including air compressibility is needed. Here, we use a numerical model based

on time-averaged Navier-Stokes equations for compressible flows (Johnson, 1992). The numerical solution is obtained using a modified version of the rhoSimpleFoam solver from the finite volume code OpenFOAM v4.0.x (Weller et al., 1998). The solver calculates a steady state solution with a segregated approach using a SIMPLE loop, with the latter solution solved using the Reynolds Averaged Navier-Stokes equations (RANS) with a LaunderSharmaKE (Launder and Spalding, 1974) turbulence model. Nakao et al. (2014) successfully used OpenFOAM for simulating the airflow on a two-dimensional NACA (National

Advisory Committee for Aeronautics) wing profile under different attack angles.

In our case, we use a simplified three-dimensional model of the Falcon wing equipped with a Probe Measurement System, which consists of a pylon and a cylindrical canister mounted under the wing (see Fig. 1). The tube of the CAS with the passive inlet was not modelled since preliminary simulations showed that the effect of the CAS tube on the concentrations measured by CAS is smaller than 5%. We adopt a comparatively large model domain with edge lengths of 10 times the instrument length

to minimize the effects of the domain boundaries. The model mesh comprises $8 \cdot 10^6$ elements. The dependency of the results on the number of mesh elements was tested, using different meshes (created with snappyhexmesh), until we found convergence of the results.

To separate CFD results from the statistical analyses conducted over the measured dataset we refer to the simulated velocity as $\boldsymbol{U}$. With this notation $|\boldsymbol{U}_0|$=TAS.

### 2.2.2  Particle motion

To describe particle motion, we adopt an Eulerian-Lagrangian approach: the Eulerian continuum equations are solved for the fluid phase (see Sec. 2.2.1), while Newton's equations described the particle motion determining their trajectories. We assume spherical particles with a density $\rho_p = 2.5$ gcm$^{-3}$ for dust and $\rho_p = 1$ gcm$^{-3}$ for water droplets. We use a one way-coupling, i.e. we only consider the flow action on particles. According to Elgobashi (1991), ignoring the effect of particle motion on

the flow itself (two-way coupling) and inter-particle collisions (four-way coupling) is a reasonable assumption for volumetric particle fractions smaller than $10^{-6}$. For dust particles, this corresponds to atmospheric concentrations lower than 2.5 gm$^{-3}$. This value is about two orders of magnitude larger than concentrations measured in dense desert dust (e.g., Kandler et al.,




2009; Weinzierl et al., 2009, 2011; Solomos et al., 2017) or volcanic ash layers (e.g., Barsotti et al., 2011; Poret et al., 2018). Single particle motion is resolved using a Lagrangian model where motion equations are integrated in time. The considered forces acting on a particle are the pressure gradient, the drag force, and the gravity.

### 2.2.3 Droplet distortion model

Fast changes in the airflow can modify the shape of water droplets causing droplet break-up and consequently strongly affecting the measured number concentration. To describe how the flow affects the shape of water droplets measured by OAP instruments mounted under-wing we use a droplet deformation model. A large body of research exists on droplet deformation and breakup (Rumscheidt and Mason., 1961; Rallison, 1984; Marks, 1998). The droplet dynamics is crucial for estimating the icing hazard of supercooled droplets on an aircraft wing (e.g., Tan and Papadakis, 2003). Flow changes experienced around the wing can

have important consequences especially when sampling supercooled droplets, for example in case of mixed-phase clouds. Jung et al. (2012) observed how shear could cause almost instantaneous freezing in supercooled droplets. Vargas and Feo (2010); Vargas (2012) used laboratory observations to investigate the deformation and breakup of water droplets near the leading edge of an airfoil. Droplet breakup, as effect of instability caused by shear on the droplet surface, was early studied by Pilch and Erdman (1987) and Hsiang and Faeth (1992). Different analytical models exist for describing a droplet in a uniform flow such

as the Taylor Analogy Breakup (TAB) model (O'Rourke and Amsden, 1987), Clark's model (Clark, 1988) and the Droplet Deformation and Breakup (DDB) model by Ibrahim et al. (1993). Vargas (2012) modifies the DDB model including the effect of a changing airflow. However, this model not fully agrees with the experimental data especially for particles larger than 1000 µm. Here we use a volume-of-fluid (VOF) method (Noh and Woodward, 1976) to determine droplet deformations as a function of droplet size and flight conditions ($p_s$, $T$, TAS). Droplets are initially assumed to be spherical with diameter $d_0$. Similar to

the TAB model a simplified problem is considered assuming that droplets are radially symmetric along the flow.

Sampled droplets experience a change of relative airspeed when approaching the instrument. For this reason, we simulate a transitional state where the airspeed varies from zero (still air) to its final value TAS-PAS (when the droplet is passing through the sampling area). The applied velocity values are calculated along the simulated trajectory of the droplet (of given $d_0$) and imposed as boundary conditions. Similar to Vargas (2012) we assume that the droplet does not exchange heat with its

surroundings and the only forces involved in the deformation of the droplet are viscous, pressure and surface tension forces. The numerical method relies on the solver InterFoam included in OpenFOAM. Numerical schemes for solving the flow are second-order implicit schemes both in the spatial and temporal discretization (Rhie and Chow, 1982). The Courant number, i.e. the flow speed multiplied by time resolution and divided by space resolution, is limited to 0.8 globally and to 0.2 at the interface, and the domain size is ten times larger than the droplet, as suggested by Yang et al. (2017).

The simulations have been performed with a water to air density ratio of 1000:1. Surface tension decreases with temperature from $\sigma$=0.75 Nm$^{-1}$ at $T$=278 K to $\sigma$ = 0.7 Nm$^{-1}$ at $T$=305 K (Vargaftik et al., 1983). The effect of a change in droplet surface tension due to the presence of impurities is not considered which seems to be a reasonable assumption given that salts increase the surface tension of seawater only by less than 1 % (Nayar et al., 2014).





## 3  Results and discussion

### 3.1  Airflow distorsion and particle concentration

Aerosol concentrations are usually expressed as particle number (or mass) per unit volume. Since the aerosol particles are embedded in the air, and the air density depends on pressure and temperature, the aerosol concentration depends as well on these. Therefore, sampling conditions, e.g. the flight level pressure or the flow-induced pressure distortion at the measurement location, influence the concentration measurement directly.

### 3.1.1  Measured and simulated airflow

To understand the effect of different flight conditions on the measurements, we selected 11 test cases (see Tab. 2) with initial data ($p_s$, $T$, TAS) chosen from flight conditions recorded during SALTRACE. Figure 3 shows a frequency histogram of the static pressure $p_s$ and the TAS recorded by the CMET system during the SALTRACE campaign. The colored dots represent the 11 selected cases. Only certain combinations of $p_s$ and TAS represent typical flight conditions for the DLR Falcon. For example, low pressure (high altitude) is associated with higher aircraft speed (when air density is lower, the aircraft needs a higher speed to have the same lift). As an example for all test cases, we first analyze the result for the specific test case u100_p900 (TAS=100 $\mathrm{ms}^{-1}$ and $p$=900 hPa, see Tab. 2). Figure 4 shows the simulated airflow in a vertical plane around the Falcon wing mounted with the simplified probe (light gray region). The pressure (a) and the airspeed (b) are expressed as the deviation in percentage from free stream conditions. In a flight, the pressure above the wing is lower than in the free stream, while the pressure below the wing is higher, resulting in a lower airspeed at the wing-mounted probe compared to free stream conditions. Pressure and velocity changes in front of an obstacle are a function of the distance from the obstacle. In case of an incompressible flow, Stokes provided an analytical expression for the velocity field in front of a sphere. However, in the case of compressible flow, a necessary assumption for a fast airplane (TAS>150 $\mathrm{ms}^{-1}$), analytical solutions have not been found yet. Figure 5 shows the ratio between the probe and free stream conditions for pressure (a) and speed (b) as a function of the distance from the instrument head. The different colors represent a selection of test cases with different TAS (increasing from light-blue to brown). The gray round shape symbolizes the simplified instrument mounted in a canister below the aircraft wing. The pitot tube is sketched in dark gray. The location of the static ports is marked with a vertical red line. The sampling area, which is located at the same horizontal distance from the instrument head as the tip of the pitot tube, is marked with a light gray line. Contrary to the incompressible case, where the ratio U/U$_0$ is independent of the airspeed, here due to compressibility the ratio is changing with U$_0$. As visible in Fig. 5b, airspeed differences between the free stream and the instrument increase at higher TAS, and is larger than the incompressible case.

Errors in the pressure are generally due to the position of the static port. As explained by Barlow et al. (1999), for a pitot tube in a laminar flow, errors are a function of the pitot tube length and the static port distance from the tip. For CAS, static ports are located 44 mm downstream from the tip. According to Fig. 5a this difference will lead to a deviation in the pressure since the pressure is exponentially decreasing as a function of the distance from the probe head. For the considered test case u100_p900 (see Fig. 4) the numerical simulation shows a 1 % error in the $p_s$ with an estimated deviation of $U$ from the free stream airspeed



of 23 %.

Figure 6 compares the ratio of free stream conditions and conditions at the wing-mounted instrument for temperature (a), static pressure (b) and airspeed (c). Different marker colors indicate the selected test cases described in Tab. 2. The simulation results refer to the pitot tube static port location. The temperature difference between free stream conditions and the probe is

decreasing from 3.5 % to 0.5 % with increasing temperature. This effect is a response to a lower airplane speed at low altitude (see Fig. 3). Also, the trend of the pressure difference in Fig. 6b shows a similar behavior decreasing from 20 % at high to 1-2 % at low altitude. Local conditions differ from free stream also for airspeed as shown in Fig. 6c with airspeed being 25 % to 35 % lower at the probe location compared with the free stream. In this context, it is worth mentioning that a longer pitot tube, as in the case of CAPS, will reduce positional errors. Indeed, the differences between $TAS_{CMET}$ and $PAS_{CAPS}$ are only

15 % to 20 % (see Appendix, Fig. A2). The simulated conditions at the pitot tube location well represent the measured data from the SALTRACE campaign with small deviations (see Fig. 6). The systematic differences in temperature need a separate explanation. Like pressure also the temperature is changing with distance from airplane body increasing near the probe head. For this reason, temperature measurements are sensitive to the measurement location. In the CAS and CAPS instruments, the temperature sensor is installed in the back, and the temperature measurement is corrected using the Bernoulli equation.

Consequently, errors in pressure will lead to an error in the temperature. This provides a possible explanation for the 1 % difference between the measurements and the simulations (Fig. 6a). The temperature bias is probably due to a combination of static pressure bias, instrumental uncertainty, and model parametrization. Howsoever, an error of 1 % in $T$ will lead, according to Eq. 3, to a PAS error smaller than 0.5 % and thus can be considered negligible.

### 3.1.2   Simulated particle concentrations and sampling efficiency

Using the simulated flow fields, we study the interaction of the flow with the particles, as explained in section 2.2.2. For each class of particles with a different density and diameter, we release $2 \cdot 10^5$ particles upstream the wing-mounted instrument at the domain border. For each particle class, we calculate the sampling efficiency $f_{eff}$ as the ratio between particles passing through the sampling area and particles released in the free stream. These numbers are normalized by the ratio of the releasing area to the sampling area. This is done by using of a Gaussian kernel that reduces the dependency of the estimated particle

concentration from the computational grid (Silverman, 1986). Figure 7 shows an example of streamlines around the wing-mounted instrument. Contours are color-coded with pressure and streamlines with flow speed. Particles slow down in the vicinity of the probe. Due to the flow distortion caused by the overpressure streamlines are bent. This effect has been observed already by King (1984). The ability of particles to adapt to flow changes is expressed by the Stokes number. The Stokes number represents the ratio of particles response time to the characteristic fluid time scale. Particles with a small Stokes number react

immediately to flow changes and consequently follow the streamlines, as the case of submicron particles. Here, to generalize the analysis according to Israel and Rosner (1982), a modified Stokes number is used, hereafter called Stokes number, which is defined as:

$$Stk^* = \frac{\rho_p U d_p^2}{L 18 \mu_{air}} \psi(Re_p) \text{ where } Re_p = \frac{pUd_p}{(T \cdot 287.058)\mu_{air}} \qquad (4)$$





$\mu_{air}$ is the dynamic viscosity of air, and $\psi$ a correction factor as a function of particle Reynolds number ($Re_p$) varying from 1 in the laminar case to 0 in the case of fully turbulent flow (Israel and Rosner, 1982). $L$ is a characteristic fluid length, here fixed to 1 m. Figure 8 shows the sampling efficiency as a function of the Stokes number. Different symbol colors represent particle diameters whereas different line colors represent fits of sigmoid functions to the Stokes numbers of the selected test

cases. The instrument sampling efficiency $f_{eff}$ is well approximated by the sigmoid fits. In the appendix, Tab. A2 presents the sampling efficiencies of the selected test cases as a function of the particle diameter and density. For large Stokes numbers, the simulated concentration at the probe is minimally affected by the flow. For example, $f_{eff} > 95\%$ holds for particles larger than 100 μm. For small particles with less inertia, the effect caused by the flow is more evident, and it leads to a sampling efficiency of ∼75 % (test case u100_p900). This effect appears less marked for test cases at higher TAS and lower pressure, where 80 %

of particles reach the sampling area.

The change of particle inertia as a function of particle diameters plays a significant role also in the particle velocity. As King (1984) reported, particle speed $v_p$ may significantly differ from the local airspeed depending on their Stokes number. Figure 9 shows the particle speed $v_p$ normalized by PAS (a) and TAS (b) for the selected test cases. Different colors represent different particle diameters and marker thickness is a function of the TAS. For each simulated case, particle speed is calculated as an

15 average of the sampled particles. For diameters smaller than 5 μm, PAS is a reasonable approximation of particle speed. Larger particles, having a higher Stokes number, are less influenced by the air flow change due to their inertia. For this reason, as visible in Fig. 9b, the velocity $v_p$ of particles larger than 50 μm can be well approximated using the TAS with an error smaller than 10 %. From Fig. 9b it is visible that at higher TAS (see gray arrow) the normalized particle speed is lower, especially for smaller particles. This is due to the air compressibility effect on airspeed, as discussed in Fig. 5, whereas TAS minimally

affects the normalized speed of larger particles.

### 3.1.3 Compressibility effect on concentration: a correction strategy

The PAS is lower than TAS (during SALTRACE, PAS/TAS≃ 70 %, see Fig. 6c). Thus, for a given number of particle counts per time interval, particle number concentrations, calculated using PAS as a reference speed, are larger than values obtained using TAS. Furthermore, the pressure and the temperature at the probe are higher than at free stream as shown in Fig. 6a and

25 b. Wrong temperature and pressure values will lead to errors of the concentration values after conversion to other conditions, e.g. those at free stream. A higher pressure value leads to a lower calculated concentration, whereas it is directly proportional to the temperature value used.

The main idea of our concentration correction strategy is to express the sampling efficiency $f_{eff}$ as a function of the Stokes number and a parameter $\alpha$ describing the difference between the probe and the free stream conditions:

$$30 \quad \alpha = \frac{p_{free}}{p_{probe}} \frac{T_{probe}}{T_{free}} \frac{TAS}{PAS} \tag{5}$$

Using $\alpha$ as variable, the sampling efficiency $f_{eff}$ can be approximated with the equation:

$$f_{eff}(\alpha, Stk) = k_1 \frac{100 - k_1}{1 + e^{-k_0(\log(Stk) - x_0)}} \tag{6}$$





The sampling efficiency values used for the fits were calculated for different flight conditions and for different distances from the probe. The coefficients are obtained by linear regression on $\alpha$: $x_0 = 0.009\alpha$ -5.46, $k_0 = 0.003\alpha + 2.157$ and $k_1 = 0.87\alpha + 11.5$. Equation 6 allows correcting particle concentrations as a function of the Stokes number and flight conditions. For each particle diameter, the first step of the correction is to estimate the corresponding Stokes number using free stream conditions ($p_s$, $T$,

TAS) and a range of particle densities. Secondly, the sampling efficiency $f_{eff}$ is calculated using the Eq. 6. For the calculation of $\alpha$ also the probe conditions are needed. Finally, particle concentrations calculated using free stream conditions are corrected for each diameter by division by the corresponding sampling efficiency value $f_{eff}$.

Fig. 10 compares estimated sampling efficiencies with sampling efficiencies obtained directly from the simulations. Two different estimation methods are considered to illustrate the benefit of our new correction strategy. In the 'Old Method' (Fig. 10a)

10 concentrations are calculated using PAS as reference speed, which are then corrected with an adiabatic expansion between the probe and free stream conditions. In contrast, the 'New Method' (Fig. 10b) uses TAS as reference speed and the fitted sampling efficiency $f_{eff}$ from Eq. 6.

The concentrations calculated with the Old Method are correct for describing the behavior of small particles (see Fig. 10a). Small particles exhibit enough mobility to be considered acting like the flow. By contrast, using probe conditions and an adi-

15 abatic expansion overestimates the particle number concentration by up to 25 % for coarse mode particles ($d_p$>2 μm). This difference will grow even larger if airspeed deviates more from free stream conditions.

The New Method (Fig. 10b), based on our correction strategy, shows good agreement, with deviations smaller than 2 % for the complete size range. The New Method also has the advantage of reducing errors that otherwise will depend on the particle diameter.

Weigel et al. (2016) provides a more rough estimation based on the concept that the air compressibility effect will cause particle accumulation near the instrument. However, the concentration at the wing instrument is apparently larger only because particles are slowed down and stay longer in the corresponding region (see Fig. 9). OPCs measure particle flux, by counting single particles passing through the sampling area, hence they are not influenced by the mentioned compressibility effect. This effect is influencing other aerosol instruments measuring directly bulk concentration such as the Hot-wire or depending on it,

as the case of the Back-scatter Cloud Probe (BCP).

### 3.1.4 Reducing positional errors for OAP

OAP size range mostly covers particles larger than 10 μm. As shown in Fig. 9 particle speed in this range is minimally affected by the flow (see Fig. 9b). Thus, TAS is a good approximation for the OAP instrument reference speed, for calculating the time frame and therefore reducing images distortion errors. Using the PAS as a reference speed for the OAP will result in images

flattened along the flow direction, with a relative error proportional to the offset from the free stream. The main idea is to use the pitot tube to measure free stream conditions by calibrating the pressure sensors such that positional errors are minimized. Similar to the analysis shown in Fig. 2, CAPS pitot tube was calibrated using the free stream conditions collected by the CMET system: a polynomial fit between free stream conditions and instrument ones is used for $q_c$ and $p_s$ collected during some test





flights[2]. A similar analysis is conducted using NASA DC-8 data provided by the MMS during the ATom missions.

Figure 11 shows the ratio of airspeed measured with the wing-mounted probe and the free conditions during the NASA DC-8 campaigns ATom-1 (a), ATom-2 (b), and the Falcon campaign A-LIFE (c). During A-LIFE and ATom-2 the pitot tube probe was calibrated to match free stream conditions. The obtained airspeed, named hereafter $TAS_{CAPS}$, shows on average a 2 %

deviation from the $TAS_{CMET}$ and 3 % from $TAS_{MMS}$. Contrary, during ATom-1, the uncorrected $PAS_{CAPS}$ shows an offset with the $TAS_{MMS}$ larger than 15 % (see Fig. 11a).

In the following we evaluate OAP image distortions along the flow to estimate the correctness of the assumed particle speed $v_p$. Since ice crystals mostly present irregular shapes, we limit our analysis to liquid droplet images.

## 3.2 Droplet deformation

The recorded shape of droplets using an OAP is a combination of the real particle shape influenced by the sampling conditions and instrumental errors such as those resulting from the wrong speed setting. A wrong speed setting will lead to a wrong frequency of image recording, for example using a lower speed results in squeezed images along the air flow direction, as discussed above.

Figure 12 shows a sequence of gray-scale images taken with the CIP in a cloud passage during the A-LIFE campaign. Pressure

and temperature conditions ($p$=840 hPa, $T$=298 K) recorded during the passage ensure droplets being in a liquid state. Image colors are the three levels of shadow recording on each photo-detector. The vertical scale is 62 pixels of 15 μm while the horizontal axis represents the timeline. Images were taken using the $TAS_{CAPS}$ as reference speed. $TAS_{CAPS}$ is obtained as explained in section 3.1.4. Most of the droplets show a spherical shape, whereas some larger droplets are deformed, with the deformation being more visible for the larger droplets. The red contours highlight droplet breakup and the blue ones indicate

large and deformed droplets that are not fully recorded.

To better understand the droplet deformation and the flow deviation from the free stream conditions we perform a statistical analysis of the images shown in Fig. 12. Figure 13 compares the deformation ratio, defined as the ratio between main droplet axes $d_y/d_x$ for the different droplet images. Images are from a selected flight sequence where we encountered liquid droplets (black markers). Following Korolev (2007), we choose droplet images only showing a small Fresnel effect and entirely con-

tained in the field of view of the CIP. Image analyses were conducted using the image processing library OpenCV (Bradski, 2000). Error bars in both directions are calculated according to the CIP size resolution of one pixel, corresponding to 15 μm and solid lines indicate the mean value of each campaign.

To extend our analysis, we included two datasets collected during the ATom campaign. The first (red) was taken during ATom-1 with the $PAS_{CAPS}$ set as reference speed for particles. For the second one (gray), recorded during ATom-2, the TAS was

estimated using the calibrated speed as shown in Fig. 11. In the case of ATom-1 (red markers), the use of the PAS as an approx-

---

[2]For DMT's instruments, like the CAPS, the pitot tube calibration can be done modifying in the PADS acquisition software the file "config.ini" with the coefficients obtained with the polynomial fit. Since in PADS temperature measurements are derived using the Bernoulli equation, reported values depend on the dynamic pressure. Consequently, the temperature values need to be recalculated during the post-processing, using the dynamic pressure at the probe obtained by inverting the fit coefficients.





imation of particle speed causes a squeezing effect along the flow direction ($d_y/d_x > 1$ for most droplets). Contrary, during ATom-2 and A-LIFE $d_y/d_x$ is more evenly distributed around 1 illustating the benefit of using TAS$_{CAPS}$ as reference speed. For small droplets (<150 μm) the large scattering is due to the limited instrument resolution (see error bars). For larger droplets error bars expressing the instrumental resolution cannot explain the data scattering, especially in case of the A-LIFE dataset.

The scattering cannot either be explained only by airspeed errors. One possible explanation can be the instability effect on the surface of the droplets, as presented in Szakall et al. (2009), and discussed below. To better understand these phenomena and estimate errors committed when using OAP images to approximate droplet volume, we extend our results using a droplet deformation model.

### 3.2.1 Quantification of droplet deformation

Before analyzing the results of the model, we want to test the numerical results by comparison with data from the experimental work of Vargas (2012). As a benchmark, we selected one of the experiments of Vargas (2012) where they observed a 1032 μm water droplet approaching an aircraft wing. The selected experiment consisted on a rotating arm, with an attached wing profile, rotating at 90 ms$^{-1}$. In Figure 14 selected images at different instants from the Vargas (2012) experiment (upper panels) are compared with the corresponding simulation results (lower panels). Since the imposed flow velocity is changing with time, the

lower panel also shows the corresponding speed values $U_{rel}$ defined as the droplet relative speed with respect to the airflow. To have a more explicit comparison, Fig. 14b shows the length of the droplet's axes of the experimental data (dots) and those simulated (lines) as a function of time and relative speed $U_{rel}$. Principal axes of individual images are determined using the length of the side of a circumscribing rectangle (as sketched in the Fig. 14). As visible in the upper panel, when the droplet approaches the airfoil $U_{rel}$ increases and the droplet starts to be squeezed along the flow direction, until the breakup process

occurs at the droplet edges for $U_{rel} \sim 60$ ms$^{-1}$. The model reproduces qualitatively and quantitatively well, over time, the behavior of the droplet with only small deviations. To extend our result to different droplet diameters and flight conditions, we use the Weber number defined as:

$$We = \frac{d \rho_{air} U_{rel}^2}{\sigma}. \tag{7}$$

$\sigma$ is the surface tension and $d$ the droplet diameter. In our case, $U_{rel}$ is changing with time from zero when the droplet is in

still air, to its maximum value when the droplet is recorded ($U_{rel}$ = TAS-PAS for large Stokes numbers). The Weber number represents the ratio of the aerodynamic forces to the surface tension forces.

To understand the data deviation found in Fig. 13 we simulated different droplet diameters. The results are shown in Fig. 15 where the deformation ratio is plotted for the simulated droplets as a function of the Weber number. Different marker colors represent different test cases where we varied droplet diameters. As a droplet approaches the airfoil, the relative speed $U_{rel}$

increases and therefore the Weber number also increases. The mechanism of droplet deformation and breakup is governed by an interplay of aerodynamic, tension and viscous forces. The distortion is primarily caused by the aerodynamic forces, whereas the surface tension and viscous forces, respectively, resist and delay deformation of the droplet. Gravitational forces play a minor role since the ratio of aerodynamic forces over gravitational forces $\rho_{air} U_{rel}^2 / \rho_d g d$, is much larger than unity.



When inertial forces grow larger than the surface tension forces, they deform the droplet causing in the worst case a breakup of the droplet by aerodynamic shattering (Craig et al., 2013). For a droplet approaching an airfoil, the viscous forces are smaller than aerodynamic and surface tension forces and the droplet breakup process is mainly controlled by We. Howarth (1963) and Prandtl (1952) showed that a droplet requires a critical Weber number ($We_{crit}$) for breakup. Wierzba (1990) studied the

critical Weber numbers (Weber number at the time of droplet breakup) when droplets interact with an instantaneous airflow in a horizontal wind tunnel. Kennedy and Roberts (1990) studied the breakup of droplets subject to an accelerating flow in a vertical wind tunnel.

The critical We for the breakup from different experimental studies with uniform airflow varies around $11 \pm 2$. Craig et al. (2013) also assumed $We_{crit} = 12$ for determining the droplet critical diameter for aerodynamic shattering on an inlet. For a

droplet approaching an airfoil, since $U_{rel}$ is changing, droplet breakup occurs at larger We compared with the case of a uniform airflow. On the other hand, the rapid change in the flow creates instability and droplets show a deformed shape already at We$\sim$ 1 (see Fig. 15). Garcia-Magariño et al. (2018) characterized the $We_{crit}$ providing an analytical equation:

$$We_{crit} = 17.5 + 17.9\tau \text{ where } \tau = \frac{\sqrt{(\rho d^3 \pi / (6\sigma))}}{U_{rel}/\frac{\partial U_{rel}}{\partial t}} \tag{8}$$

Using the simulated field and Eq. 8 with Eq. 7, we can express the critical diameter $d_{crit}$ as a function of relative speed $U_{rel}$.

$U_{rel}$ is a function of TAS and mounting position (see Fig. 5). Therefore, for a specific configuration $d_{crit}$ can be expressed as a function of TAS. Figure 16 shows how $d_{crit}$ decreases when TAS increases. The two colors refer to the two different mounting configurations for the DLR Falcon and the NASA-DC-8. The differences in the mounting system between the Falcon and the DC-8 are responsible for the differences in the relative velocity. Generally, a large difference between the free stream and the probe airspeed will increase the relative velocity and consequently the Weber number, reducing the critical diameter for

droplet breakup. This explains why critical diameters for the Falcon configuration are smaller than for the DC-8. Small droplets resist deformation more than larger ones because the small radius translates into a larger curvature. However, when comparing equal Weber numbers, small droplets show instability phenomena where the droplet surface starts to oscillate (called Taylor instability). This effect can be responsible for the scattering of data in Fig. 13. The deforming effect is only partially visible in the statistical analysis presented in Fig. 12, since large particles have a higher chance to be only partially recorded inside the

field of view, and consequently being excluded from the study. As shown in Fig. 5, the relative velocity $U_{rel}$ at the sampling area location is roughly 30 % of TAS. For example, for an airplane flying at $100 \text{ ms}^{-1}$ at 900 hPa, recorded particles have a relative velocity of $30 \text{ ms}^{-1}$. For a droplet of 200 μm, this corresponds to We=2.5 and consequently it cannot be considered being spherical.

### 3.2.2   Impact of droplet deformation on particle volume estimation

As observed in Figs. 12, 14 and 15 large liquid droplets show a large distortion, when measured with an OAP onboard a fast aircraft. This raises the question of which diameter should be used to describe droplets. Different diameter definitions exist (Korolev et al., 1998). Here we use as approximation diameters $d_{approx}$, the maximum diameter $d_{max} = \max(d_x, d_y)$, the mean diameter $d_{mean} = (d_x + d_y)/2$, and the area equivalent diameter $d_{equi} = 2\sqrt{Area/\pi}$ where Area is the droplet cross sec-





tion area calculated from the image. Two further approximation are used, $d_{spheroid} = (d_x d_y^2)^{1/3}$ derived assuming a spheroid rotated around the x-axis and $d_{asym} = (4/\pi \cdot Area \cdot d_y)^{1/3}$. McFarquhar (2004) noted that inconsistencies in particle size definitions could have significant impacts on mass conversion ratios between different hydrometeor classes used in numerical models. Errors in the droplet volume approximations have a direct effect on the LWC estimation. Here, we use the numerical

simulations analyzed in Fig. 15 to better understand possible errors in the estimation of the droplet volume. The simulated droplet shapes are processed to calculate the chosen different approximation diameters and estimate the corresponding approximation volumes ($V_{approx} = d_{approx}^3 \cdot \pi/6$). Results are shown in Fig. 17, where the relative error ($err_{rel} = |V_{approx} - V_0|/V_0$) is plotted as a function of We. $V_0$ is the initial droplet volume. Using the $d_{max}$ (red diamonds) error increases progressively with We, from 10 % at We=1 until almost a factor 10 when the breakup process starts. A better approximation, reducing this

effect, is to use the mean diameter $d_{mean}$ (cyan symbols). In this case, for We<20, on average the volume is underestimated by 2 % to 20 %. For larger We, the formula overestimates the volume up to a factor of 6. A more stable way to define droplet diameter is based on the equivalent $d_{equi}$ (green circles). Also in this case, errors are growing as a function of We, passing from 3 % to 40 %. A common assumption is considering droplets as spheroids (purple triangles). In this case using the approximation formula for a spheroid $V_{approx} = \pi/6 d_x \cdot d_y^2$ gives errors smaller than 12 % below We=34. For larger We, droplets

appear asymmetric, and errors can grow larger than a factor of 7. The best approximation is obtained by using $d_{asym}$ where the volume result in the more general formula $V_{approx} = 2/3 Area \cdot d_y$. Errors, in this case, are generally $\pm 3\%$ to $\pm 10\%$ and in case of droplet break up still smaller than a factor of 2.

## 4  Recommendations

The following list summarizes the proposed correction strategy to reduce flow-induced measurement errors and to express measurement uncertainties for OAP and OPC instruments. OPC and OAP measurement errors directly depend on flow conditions like pressure, airspeed, and temperature. Since free stream conditions differ from conditions at the position where the instrument is mounted on the aircraft, it is fundamental to adopt a correction scheme.

Recommended steps for OAP:

– For imaging probes, covering a size range larger than 50 μm, use the TAS as the reference particle speed for calculating the shutter speed. If possible, use the TAS recorded by the plane. Otherwise, an option could be to re-calibrate the pitot tube installed on the probe to measure free stream conditions (see Sec. 3.1.4 and the appendix). In this last case, the probe conditions can be obtained during the data evaluation by inverting the calibration coefficients for $p_s, q_c$ and using Eq. 3 to calculate the PAS.

– For approximating the droplet volume using OAP images, use the formula $V = 2 Area \cdot d_y/3$. Indeed, droplets appear to be deformed by the flow distortion around the wing-mounted instrument even at low airspeed.





- – Try to find instrument mounting locations where the deviation between the instrument and free stream conditions are small. Substantial velocity differences will lead to deformations of droplets with the risk of aerodynamic shattering, and consequently increasing the number concentrations. See Fig. 16 for a quick-view of critical breakup diameters depending on aircraft speed.

Recommended steps for OPC:

- – As a first step calculate the $\alpha$ parameter using the ratio between the free stream and probe flight conditions.

- – Second, estimate the particle Stokes number based on flight conditions ($p$,TAS), particle diameter and density (Eq. 4). If density is not known, use a range of possible values to propagate the uncertainty.

- – Use the Eq. 6 function of $\alpha$ and $Stk$ to calculate the correction factor. This parameter depends on flight conditions and
particle diameter.

- – Use free stream conditions (TAS) to calculate particle concentrations. This is especially important for coarse mode aerosols.

- – Divide the obtained concentrations by the correction factors.

- – The first steps can be skipped using the lookup table that can be found in the appendix (Tab. A2). These correction values
were calculated for different diameters and two reference densities (water and dust).

## 5  Conclusions

This study investigated the effect of flow distortion around wing-mounted instruments. The analysis focused on open path and passive inlet OPC and OAP instruments. The data-set collected during SALTRACE (Weinzierl et al., 2017) was used to esti-mate flow differences between the free stream and the aerosol and cloud probes mounted under an aircraft wing. The airspeed
at the probe location (PAS) was on average 30 % smaller than at free stream conditions (TAS).
A CFD model was adopted to test different flight conditions. The numerical results matched the recorded difference between free stream conditions and the conditions at the probe location (see Fig. 6). The simulated flow field was used to estimate changes in concentration for particles of different density and diameters. Concentrations of particles smaller than 0.5 µm were correctly estimated when using probe conditions (PAS). However, simulations showed that using probe conditions such as
PAS, $p_{probe}$ and $T_{probe}$ led to incorrect particle concentration with an overestimation of the coarse mode aerosol amount of up to 20 % especially in the range $[5 - 100]$ µm (see Fig. 10). We proposed a correction factor for the particle concentration based on the particle Stokes number. This correction was generalized to different configurations with a simple formula based on the ratio between the probe and free stream conditions (Eq. 6), reducing concentration errors drastically, from 30 % to smaller than 2 %.
Wrong OAP recording speeds result in deformed images. The deviation of the airspeed recorded by the under-wing instrument





from the free stream airspeed was significantly reduced after a re-calibration of the instrument's pitot tube. The re-calibrated speed was used as a reference for the CIP during the A-LIFE and Atom-2 missions. Although the use of TAS as a reference particle speed largely reduced images distortions, large droplets still appeared deformed. To understand the deformation of water droplets, a VOF model was used. The model well reproduced experimental data from Vargas (2012). Already at We=1

droplets appeared deformed (see Figs. 14 and 15). In particular droplets smaller than 400 µm showed a wiggling behavior. Oscillations of the droplets were caused by Taylor instabilities developing at the surface.

To reduce errors of the estimated LWC, derived from OAP size distributions, different definitions of droplet diameter were tested. Using the maximum droplet dimension $d_{max}$ to estimate droplet volume resulted in a 40 % error even at low aircraft speed with errors dramatically increasing, up to one order of magnitude, with aircraft speed. The best volume approximation

was obtained by using the formula $V = 2d_y \cdot Area/3$, where $d_y$ is the particle diameter perpendicular to the flow and $Area$ is the droplet area calculated from the image. Significant differences between airspeed at the free stream (TAS) and at the instrument location (PAS) increased the risk, especially for fast flying aircraft, of the breakup of large droplets. Droplet breakup caused measurement artifacts by increasing the number of particles (red contours in Fig. 12). This phenomenon, known also as aerodynamic breakup (Craig et al., 2013), caused shattering of droplets without hiting instrument walls. Extending the re-

sult from Garcia-Magariño et al. (2018), we provided an estimate for the critical diameter of droplet breakup as a function of aircraft speed.

Numerical simulations explained the physical reason for the observed deviations between TAS and PAS and provided a correction method. Using this new method for the analysis of past and upcoming data sets may reduce errors in particle and droplet number concentrations up to 30 % and in the derived LWC up to one order of magnitude.




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





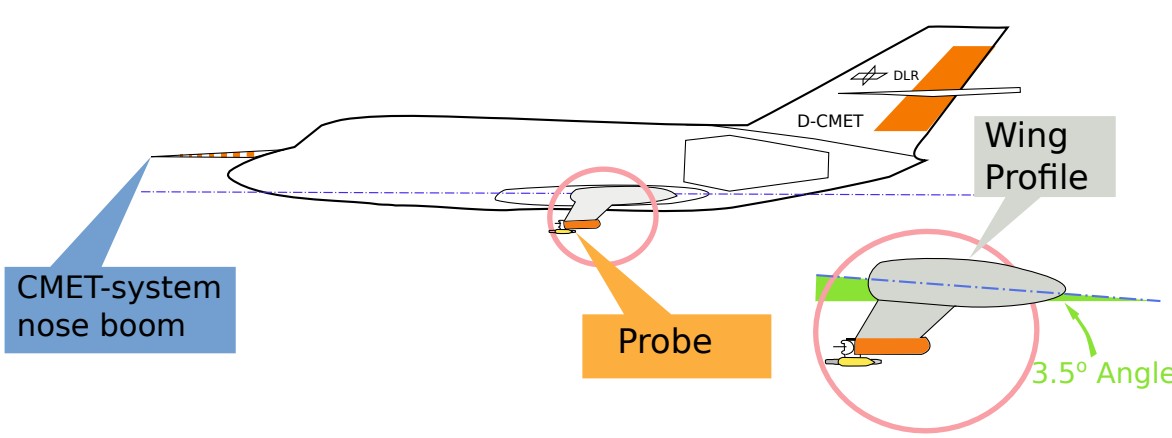

**Figure 1.** The DLR research aircraft Falcon equipped with the meteorological sensors in the nose boom (also referred to the "CMET system" in this study) and a probe mounted under the wing for the detection of coarse mode aerosols and cloud droplets/ice crystals.





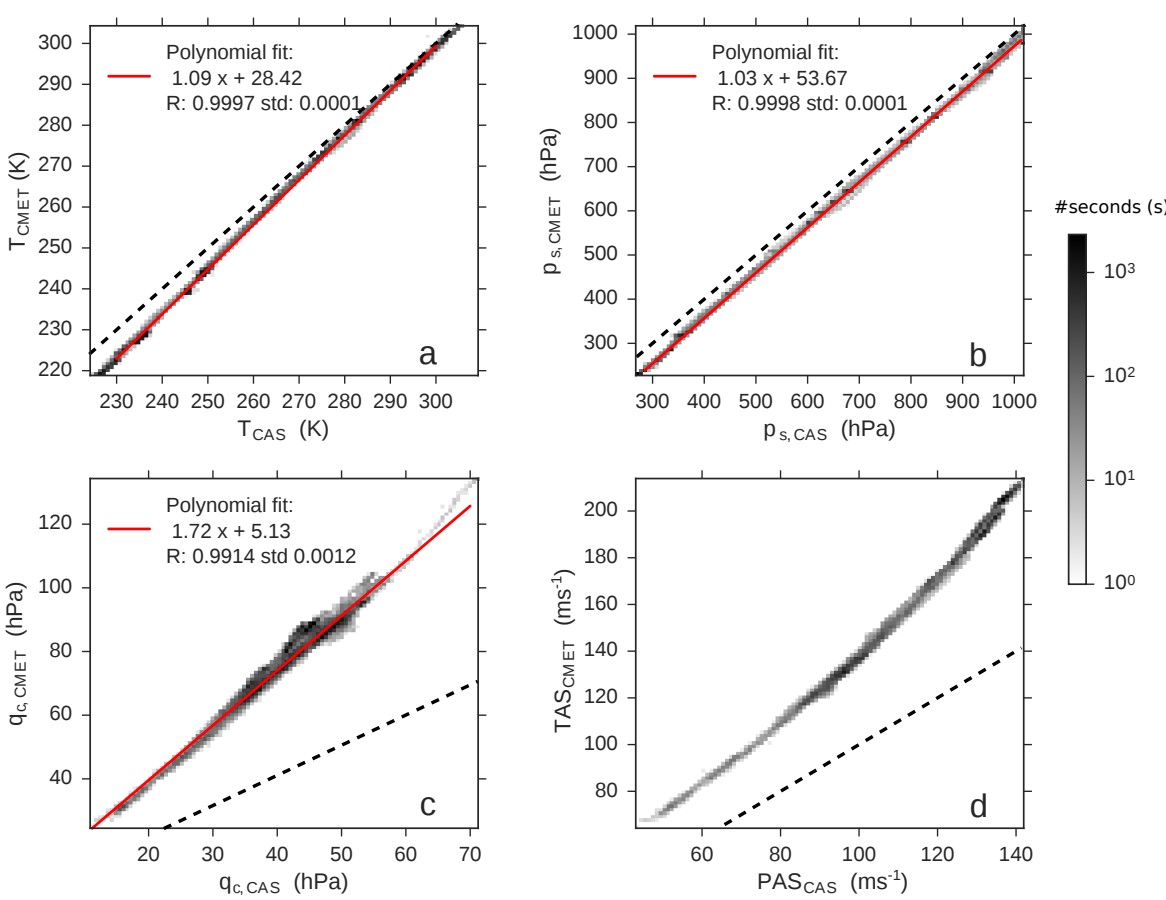

**Figure 2.** Statistical comparison between values recorded by the CMET systyem and the CAS pitot tube during SALTRACE: temperature (a), static pressure (b), dynamic pressure (c) and airspeed (d). Histogram color-map refers to the number of seconds. Dashed lines represent the 1:1 line and the red lines linear fits.





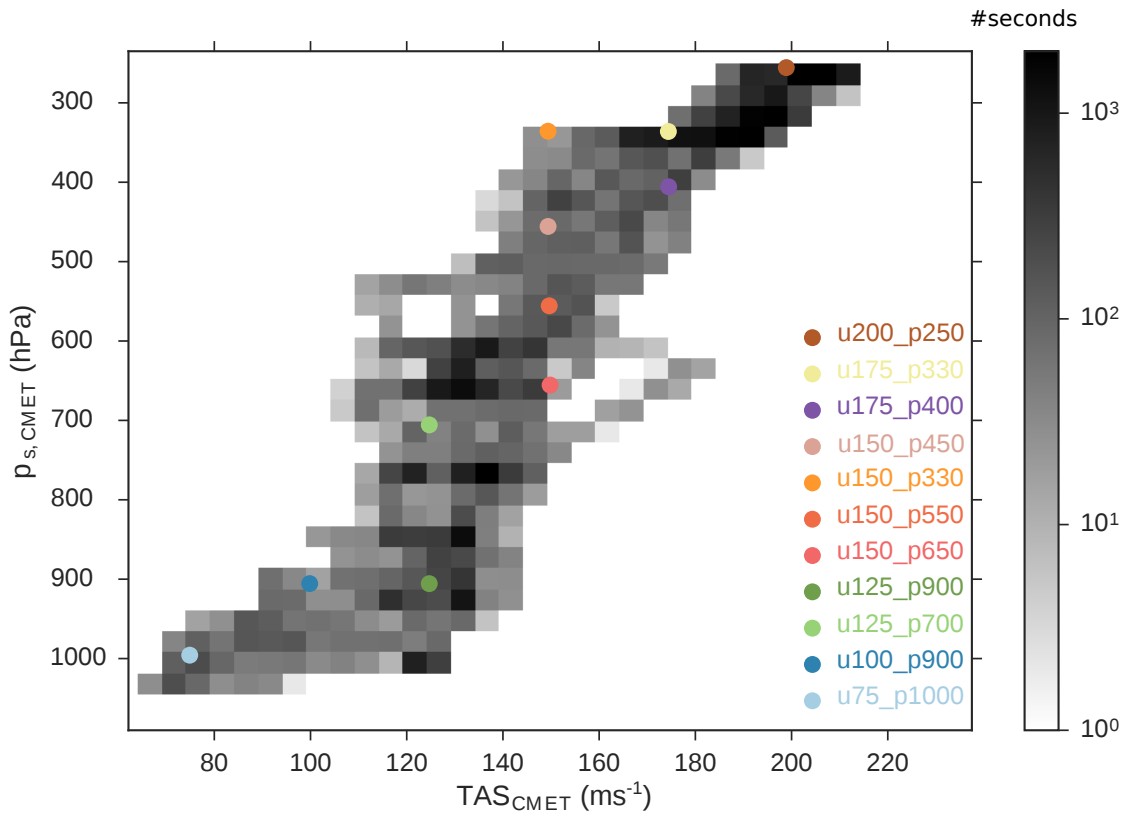

**Figure 3.** 2D histogram of flight conditions (pressure and TAS) recorded by the CMET system at the nose boom during SALTRACE. Pixels are color coded with the number of seconds spent in the corresponding condition. Colored dots represent the selected test cases for the CFD investigations described in Tab. 2.





**Figure 4.** Vertical slice through the probe center and along the flow direction for the simulated test u100_p900. The gray area represents the aircraft wing with the pylon and the probe installed. Colored contours illustrate the static pressure (a) and the velocity field (b) expressed as the ratio of the free stream values. The overpressure in front of the probe is slowing down the flow field.



**Figure 5.** Static pressure and velocity normalized by the free stream values calculated along a streamline as a function of the distance from the instrument head. Different colors represent different tests described in Tab. 2. The gray area marks the pitot tube location, while the red area marks the static port location. The pitot tube was designed to measure the pressure conditions representative for the sampling area of the instrument.





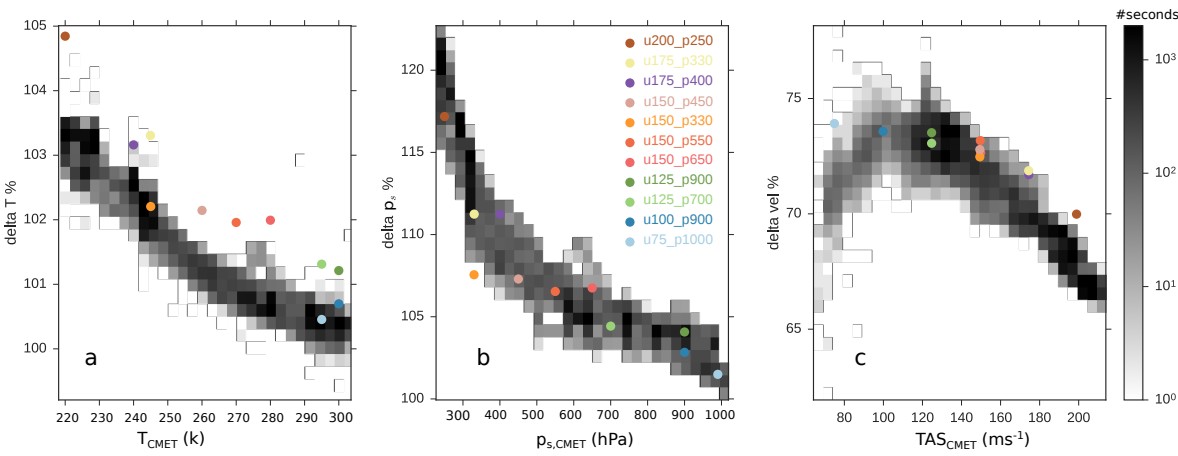

**Figure 6.** Statistical analysis of differences between values read by the CMET system and the CAS pitot tube during SALTRACE. Histogram color-map refers to the number of seconds. Colored dots represent the selected test cases for the CFD simulations described in Tab. 2.





**Figure 7.** Pressure profile around the probe for test case u100_p900. Lines represent streamlines colored with flow velocity. The wing influences the streamlines that appear bent around the probe.





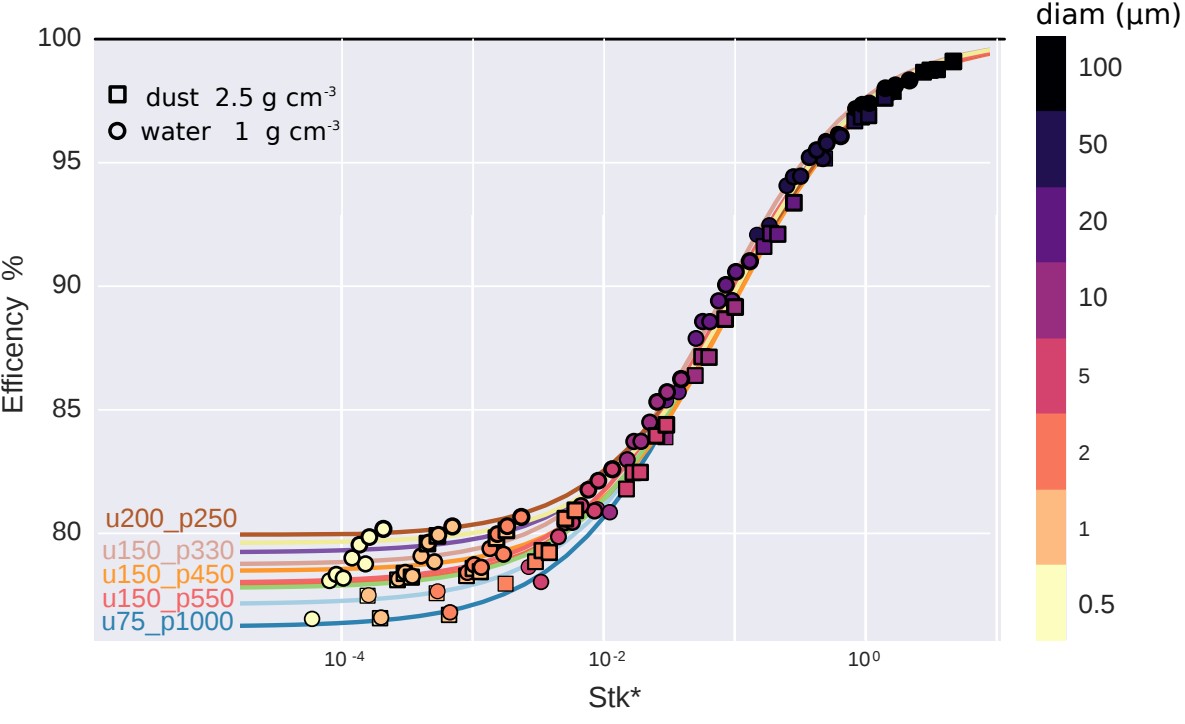

**Figure 8.** Sampling efficiency calculated as a function of modified Stokes number (see Eq. 4) for selected numerical test cases of Tab. 2. Each marker represents a run where we released $2 \cdot 10^5$ particles of a specific diameter (colors) and density (markers) in front of the probe in the computed flow field. Sampling efficiency is defined as the ratio between particles released and particles passing through the sampling area, renormalized by the corresponding areas. The black line marks 100 % efficiency. Curves are obtained by fitting the data with a sigmoid function.





**Figure 9.** Particle velocity normalized by $\text{PAS}_{CAS}$ (a) and $\text{TAS}_{CMET}$ (b) as a function of modified Stokes number $Stk^*$. Marker thickness increases with TAS. Colors denote particle diameters.





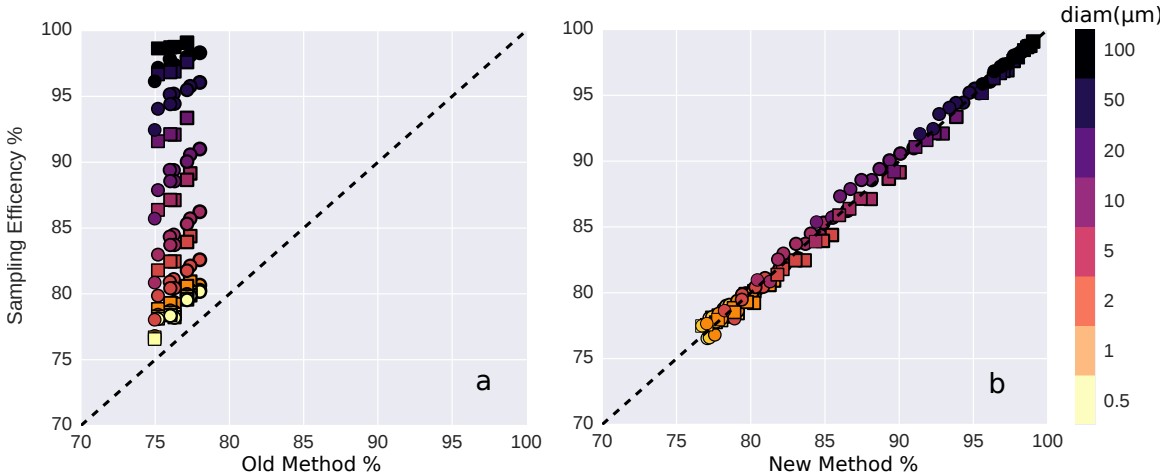

**Figure 10.** Simulated sampling efficiency versus sampling efficiencies estimated from the simulations using the "Old Method" (a) and the "New Method" (b). Different markers indicate different numerical test cases while the colors refer to the particle diameter. For the Old Method, the concentration is calculated using the probe conditions and using an adiabatic expansion to free stream conditions. For the New Method, the concentration is calculated using free stream conditions and corrected for the flow distortion by applying Eq. 6. To obtain the sampling efficiency, both concentrations are divided by concentration assumed at the free stream.





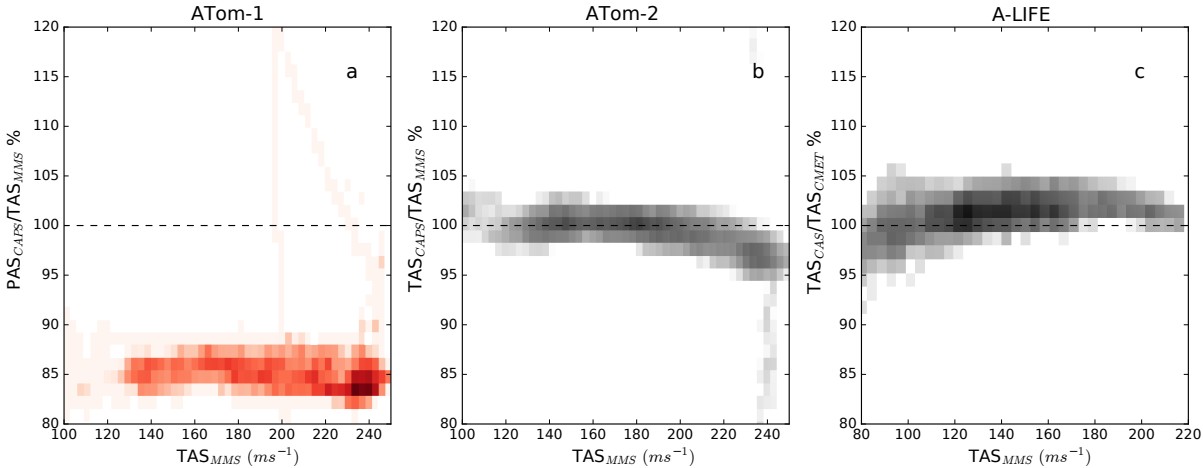

**Figure 11.** Air velocity recorded by CAPS normalized by air velocity recorded by the default aircraft systems. Data from ATom-1 (a), ATom-2 (b) and A-LIFE (c) is shown. While the CAPS pitot tube was calibrated to match PAS during ATom-1, it was calibrated to match the TAS during ATom-2 and A-LIFE. The pixels are color coded with the number of seconds.





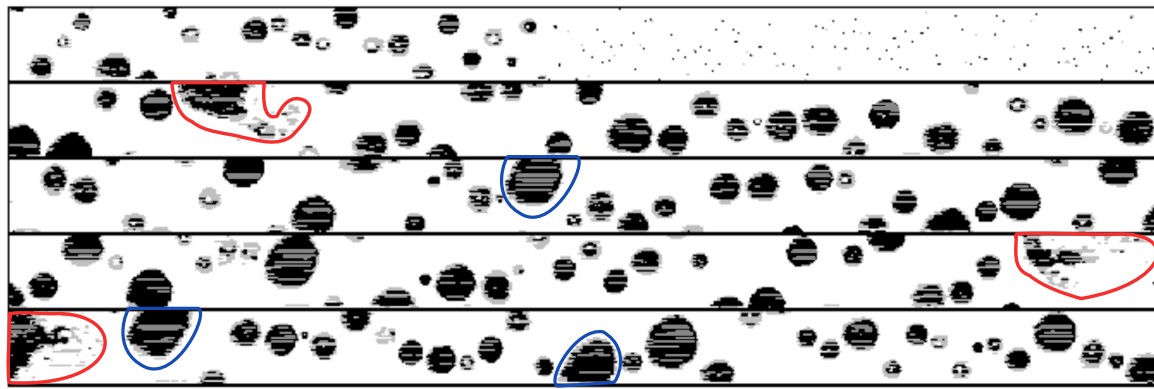

**Figure 12.** CIP gray-scale images collected in a cloud during the A-LIFE campaign. Colors are the three levels of shadow recorded on each photo-detector. The vertical scale is 62 pixels of 15 μm while the horizontal axis represents the time line. Flight conditions are TAS=124 ms$^{-1}$, $p_s$=840 hPa, and $T$=298 K. Red contours indicate splashes due to droplet breakup. Blue contours highlight large particles that are not complitely recorded.





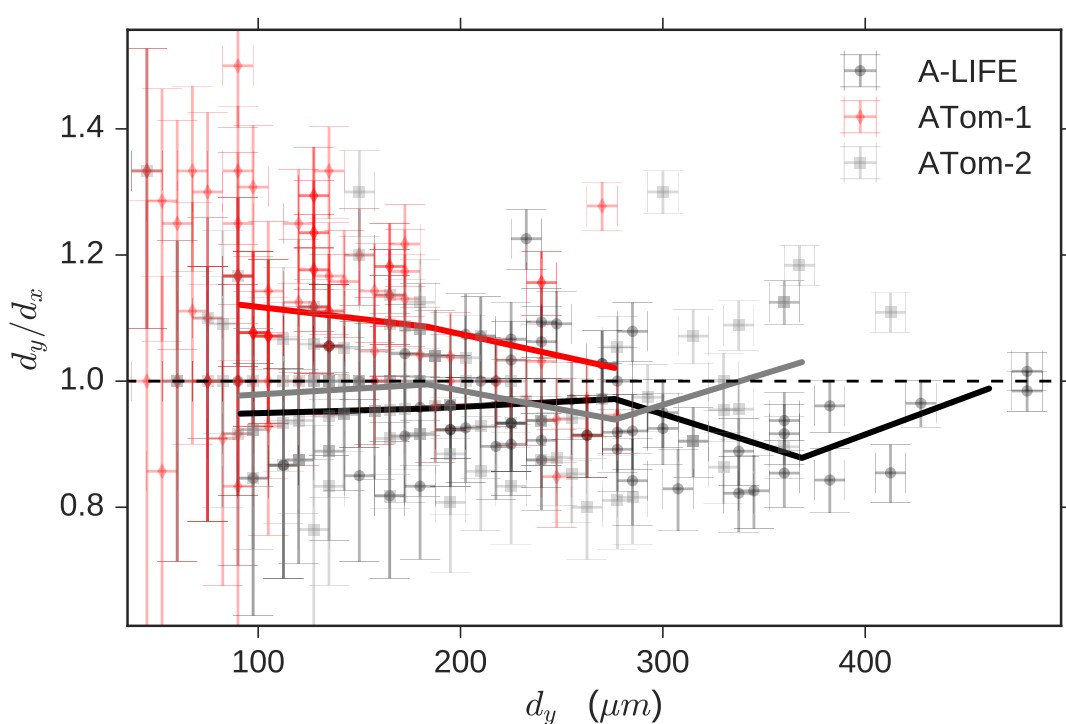

**Figure 13.** Ratio between the main axis lengths ($d_y/d_x$) of droplets recorded by the CIP during different campaigns (indicated by color). Error bars represent the CIP pixel resolution (15 µm). Considered are only particles recorded during cloud passages where the temperature ensures a liquid droplet state. Black markers represent 200 particles selected from Fig. 12. Red markers show data from the ATom-1 mission where the CAPS pitot tube, and thus the OAP reference velocity, was calibrated to match PAS whereas the data of the other campaigns (shown in gray and black) was collected when the CAPS pitot tube was calibrated to match TAS (see also Fig. 11). Lines indicate the mean values of each campaign.





**Figure 14.** Simulations for a test case with a 1032 μm diameter droplet reproducing an experiment by Vargas (2012). In the upper panel the upper halves display images recorded by Vargas (2012) while the lower halves show corresponding simulated droplets. The air flow comes from the left. Time and relative airspeed increase from the left to the right. The lower panel shows changes of both droplet axes lengths ($d_x$ and $d_y$, see inlay image) as a function of time and relative velocity (labeled at the top) for the experiment from Vargas (2012) (dots) and for the simulations (continuous line).





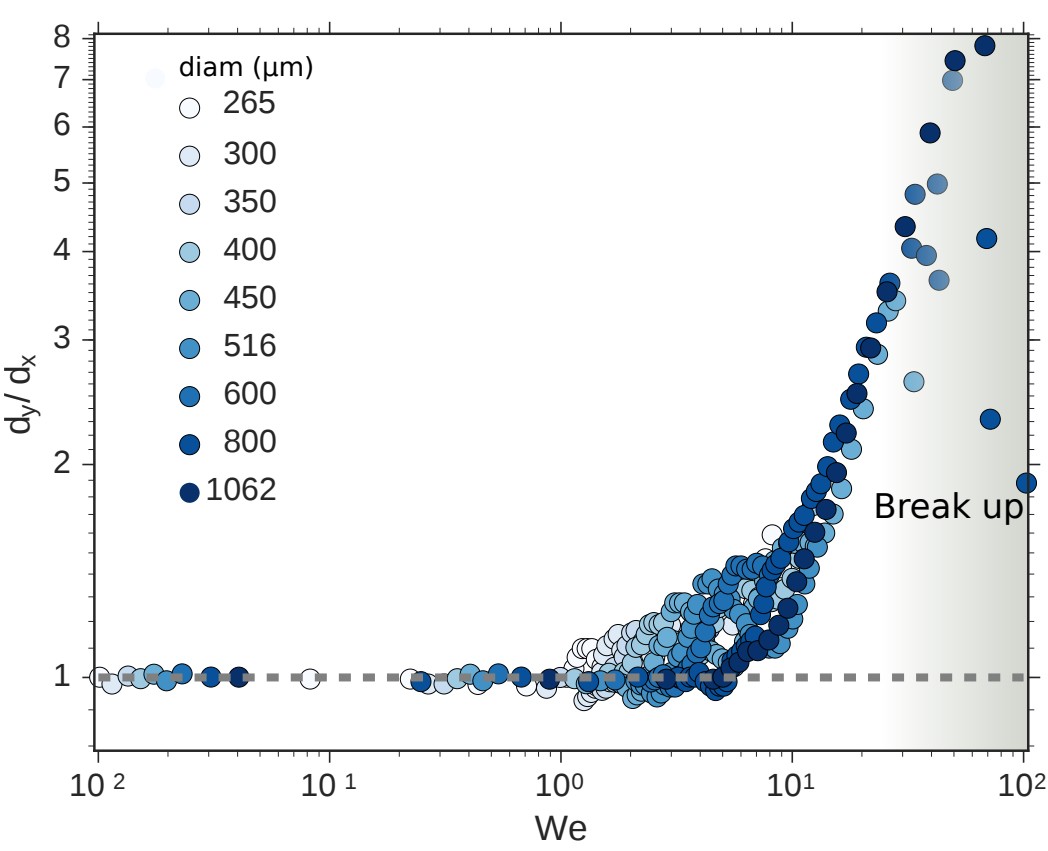

**Figure 15.** Droplet axis ratio ($d_y/d_x$) as a function of Weber number (We). Different dots represent different simulations where we increased the droplet diameter from 265 μm (white) to 1062 μm (dark blue). The gray area represents the region where droplet break up can occur.





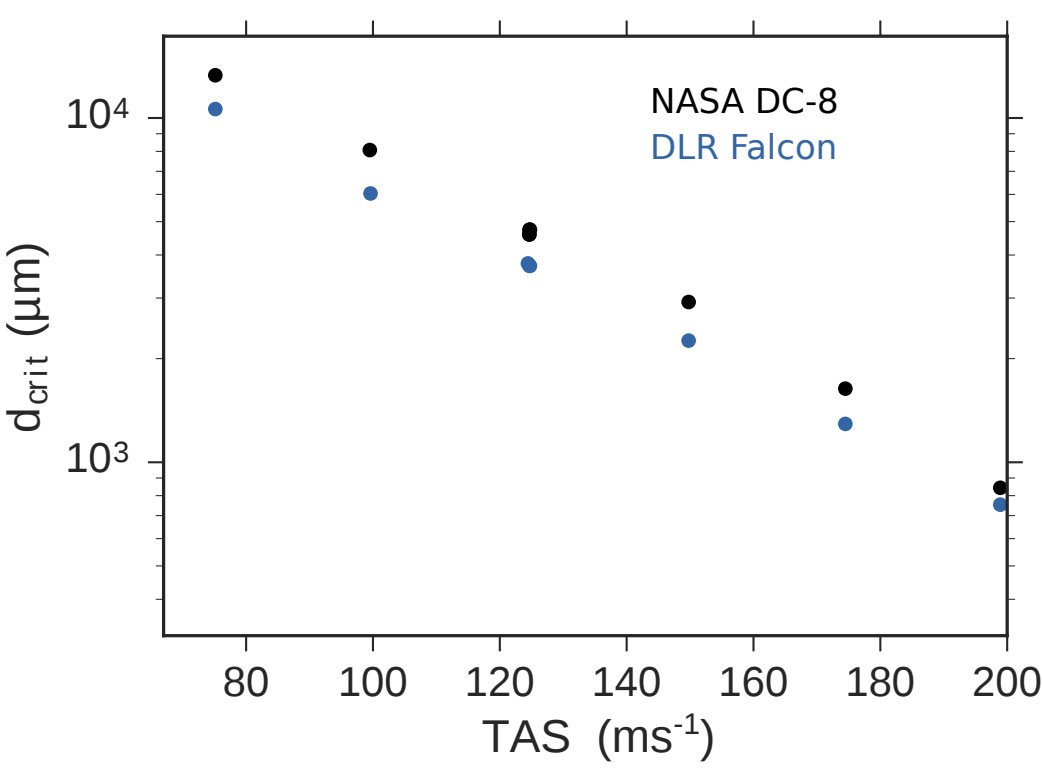

**Figure 16.** Critical diameter for droplet breakup as a function of TAS for the DLR Falcon and the NASA DC-8 configuration.





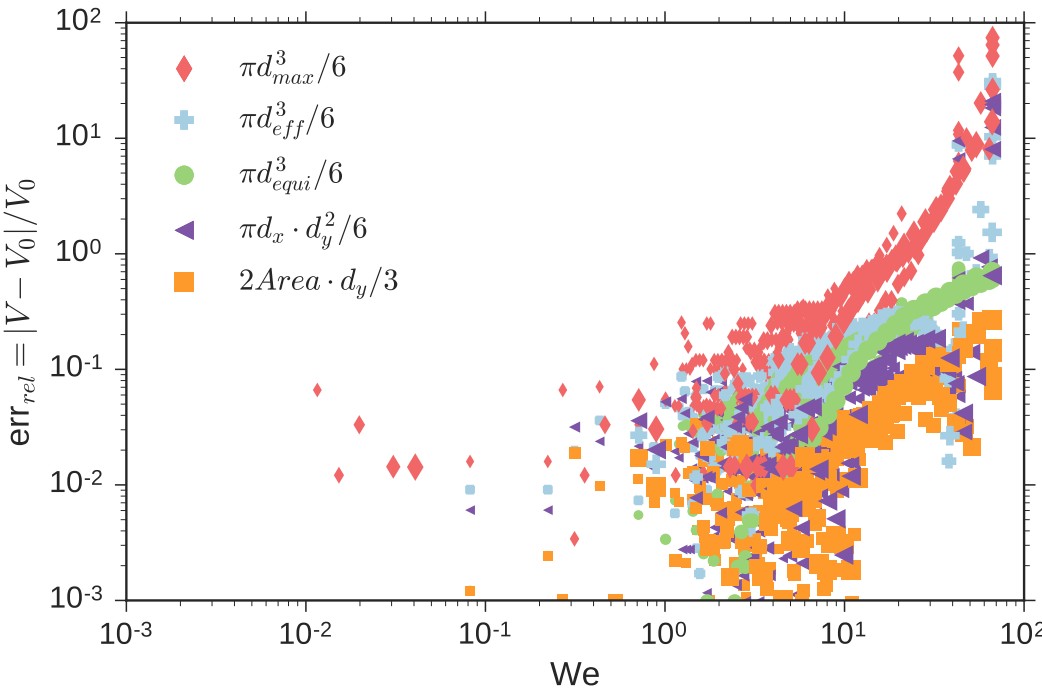

**Figure 17.** Relative error of the droplet volume as a function of Weber number (We) for different volume approximations (colors). $V_0$ is the original droplet volume. Different marker sizes represent different simulations where we varied the droplet diameter.



**Table 1.** Airplane configuration details and instruments used in this study. The DLR CAS (UNIVIE CAPS) is equipped with a 17 cm (24 cm) long pitot tube.

| Campaign | Airplane | Max. altitude | Typ. cruise speed | Default flow sensors | Wing instrument | Reference |
|----------|----------|---------------|-------------------|----------------------|------------------|-----------|
| SALTRACE | DLR Falcon | 12800 m | 80-220 $\text{ms}^{-1}$ | CMET system | DLR CAS | Weinzierl et al. (2017) |
| A-LIFE | DLR Falcon | 12800 m | 80-220 $\text{ms}^{-1}$ | CMET system | UNIVIE CAPS | Weinzierl and ALIFE_Team (2018) |
| ATom | NASA DC-8 | 13800 m | 90-250 $\text{ms}^{-1}$ | MMS | UNIVIE CAPS | Wofsy et al. (2018) |

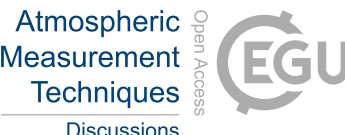

**Table 2.** Flight conditions ($p$, $T$, TAS) used to initialize the numerical flow simulation test cases.

| Test name | TAS ( $\mathrm{ms}^{-1}$) | $p$ ( hPa) | $T$ ( K) |
|---|---|---|---|
| u75_p1000 | 75 | 1000 | 300 |
| u100_p900 | 100 | 900 | 300 |
| u125_p700 | 125 | 700 | 295 |
| u125_p900 | 125 | 900 | 300 |
| u150_p650 | 150 | 650 | 280 |
| u150_p550 | 150 | 550 | 270 |
| u150_p450 | 150 | 450 | 260 |
| u150_p330 | 150 | 330 | 245 |
| u175_p400 | 175 | 400 | 240 |
| u175_p330 | 175 | 330 | 245 |
| u200_p250 | 200 | 250 | 220 |





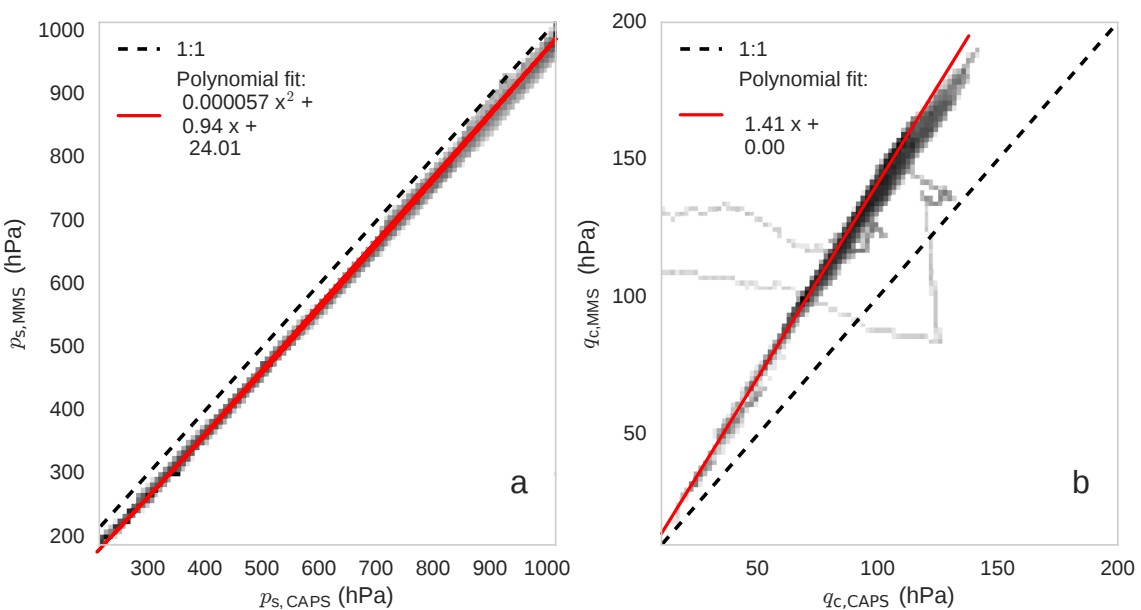

**Figure A1.** Statistical comparison between values recorded by the MMS and the CAPS pitot tube installed under the aircraft wing during ATom-1: static pressure (a) and dynamic pressure (b). The histogram color-map refers to number of seconds. Dashed lines represent the 1:1 line and red lines polynomial fits.




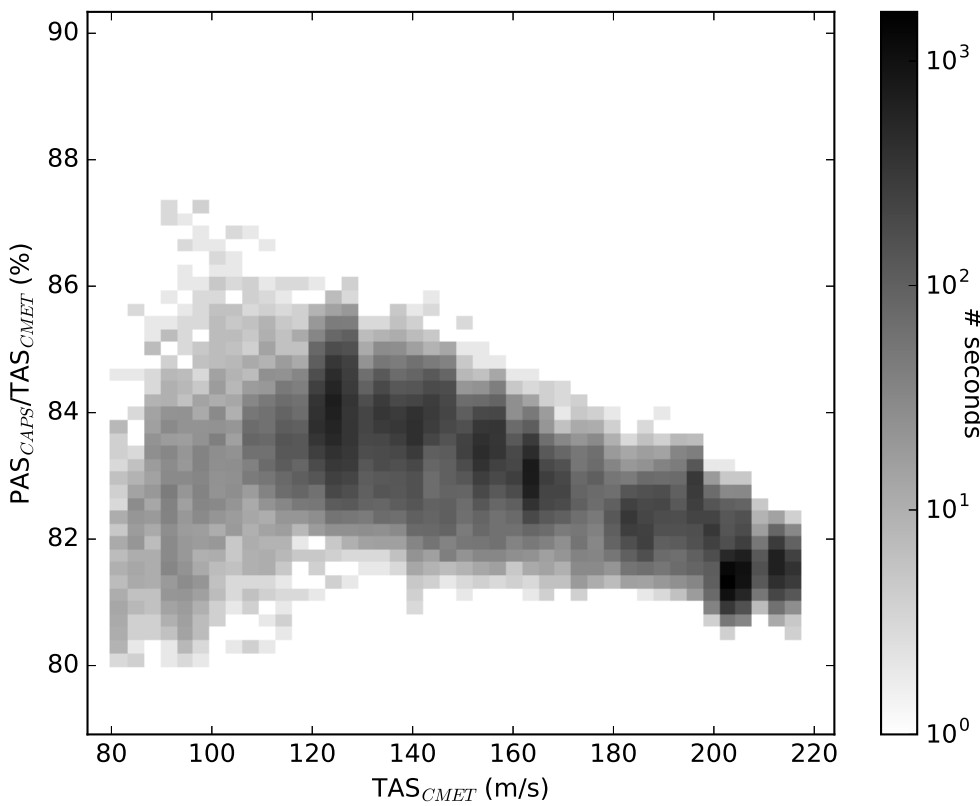

**Figure A2.** Statistical analysis of differences between airspeed at the free stream and at the probe during A-LIFE when CAPS was calibrated to match free stream conditions. $TAS_{CMET}$ values were obtained by the CMET system and the $PAS_{CAPS}$ was post-calculated using Eq. 1 and the dynamic and static pressure as well the temperature value at the probe obtained inverting the relation described in section 3.1.4. The histogram color-map refers to the number of seconds.



**Table A1.** Acronyms and symbols used.

| | |
|---|---|
| CFD | Computational Fluid Dynamic |
| LWC | Liquid Water Content |
| OAP | Optical Array Probe |
| OPC | Optical Particle Counter |
| PAS | Probe Air Speed |
| TAS | True Air Speed |
| VOF | Volume Of Fluid |
| $d$ | Particle diameter |
| $v_p$ | Particle speed |
| $V$ | Droplet Volume |
| $U$ | Flow speed |
| $\rho$ | Density |
| $\sigma$ | Surface tension |
| $\gamma$ | Heat capacity ratio |
| $p_s$ | Static pressure |
| $p_{tot}$ | Total pressure |
| $q_c$ | Dynamic pressure |
| $M$ | Mach Number |
| Stk | Modified Stokes number |
| We | Weber Number |

*Author contributions.* BW and AS designed the study. AS carried out the simulations and the numerical analysis. BW and MD performed the airborne CAPS measurements. TPB provided the MMS data. AS analyzed the results with the help of BW and wrote the manuscript with the support of BW and JG. All authors commented on the manuscript.

*Competing interests.* The authors declare that they have no competing interests.

5  *Data availability.* Data and materials availability: The data are archived in the World Data Center ORNL DAAC and can be accessed under the doi:10.3334/ornldaac/?? (Spanu et al., 2018)

*Acknowledgements.* This work was supported by the VERTIGO Marie Curie Initial Training Network, funded through the European Seventh Framework Programme (FP7 2007-2013) under Grant Agreement number 607905, and by the European Research Council under the European Community's Horizon 2020 research and innovation framework program/ERC grant agreement number 640458 (A-LIFE). The



**Table A2.** Sampling efficiency values (%) for different test cases as a function of particle diameter and density as shown in Fig. 8.

| diam (µm) | 0.5 | 1 | 2 | 5 | 10 | 20 | 50 | 100 |
|---|---|---|---|---|---|---|---|---|
| test case | | | | | | | | |
| | $\rho_p = 1 \ \mathrm{gcm}^{-3}$ (water) | | | | | | | |
| u75_p990 | 76.89 | 76.93 | 77.09 | 78.09 | 80.50 | 84.99 | 91.86 | 95.75 |
| u100_p900 | 76.60 | 76.68 | 76.82 | 78.06 | 80.89 | 85.85 | 92.35 | 96.12 |
| u125_p900 | 77.77 | 77.81 | 78.11 | 79.46 | 82.55 | 87.42 | 93.50 | 96.25 |
| u150_p650 | 78.27 | 78.42 | 78.76 | 80.43 | 83.70 | 88.50 | 94.42 | 97.08 |
| u150_p550 | 78.18 | 78.29 | 78.64 | 80.32 | 83.76 | 88.52 | 94.44 | 96.71 |
| u175_p400 | 79.42 | 79.52 | 79.87 | 81.66 | 85.20 | 90.03 | 95.57 | 98.62 |
| u175_p330 | 79.75 | 79.80 | 80.14 | 82.14 | 85.81 | 90.66 | 95.64 | 98.70 |
| u200_p250 | 80.16 | 80.30 | 80.66 | 82.60 | 86.22 | 91.12 | 96.18 | 97.04 |
| | $\rho_p = 2.5 \ \mathrm{gcm}^{-3}$ (dust) | | | | | | | |
| u75_p990 | 76.91 | 77.01 | 77.40 | 79.49 | 83.48 | 88.93 | 95.01 | 97.77 |
| u100_p900 | 76.62 | 76.71 | 78.11 | 80.48 | 84.88 | 90.13 | 95.77 | 97.71 |
| u125_p900 | 77.76 | 77.93 | 78.64 | 81.39 | 85.89 | 91.06 | 96.61 | 98.70 |
| u150_p650 | 78.34 | 78.61 | 79.28 | 82.51 | 87.13 | 92.25 | 96.23 | 99.25 |
| u150_p550 | 78.24 | 78.47 | 79.12 | 82.51 | 87.18 | 92.10 | 97.49 | 98.99 |
| u175_p400 | 79.47 | 79.74 | 80.58 | 84.02 | 88.62 | 93.61 | 98.07 | 98.94 |
| u175_p330 | 79.77 | 80.06 | 80.88 | 84.32 | 89.18 | 94.40 | 98.50 | 99.20 |
| u200_p250 | 80.26 | 80.79 | 81.66 | 85.20 | 89.85 | 94.22 | 97.97 | 99.26 |

SALTRACE aircraft field experiment was funded by the Helmholtz Association (Helmholtz-Hochschul-Nachwuchsgruppe AerCARE, Grant Agreement VH-NG-606) and by DLR. We also would like to acknowledge partial funding through LMU Munich's Institutional Strategy LMUexcellent within the framework of the German Excellence Initiative. The A-LIFE field experiment was funded under ERC grant agreement number 640458 (A-LIFE). In addition, DLR and two EUFAR projects provided funding for a significant amount of additional flight
5   hours and aircraft allocation days for the A-LIFE aircraft field experiment. The ATom mission and the MMS measurements were funded under National Aeronautics and Space Administration's Earth Venture program (grant NNX15AJ23G). CAPS measurements during ATom were additionally supported by the University of Vienna. We are thankful to Andreas Giez, Volker Dreiling, Christian Mallaun, and Martin Zoeger for providing the meteorological data for SALTRACE and A-LIFE. We would like to thank DMT for the fruitful discussions related to the CAPS instrument. Thanks to the SALTRACE, A-LIFE, and ATom science teams, the pilots, administrative and technical support teams
10  for their great support before and during the field missions and for the accomplishment of the unique research flights.