# Peer review of "Flow-induced errors in airborne in-situ measurements of aerosols and clouds"

_Atmospheric Measurement Techniques, 2019_

## Referee Comment (RC1) · Anonymous Referee #1 · 11 Mar 2019

**1   General Comments**

This manuscript is a useful addition to the growing body of work investigating the influence of airborne particle instruments on the measurements that they are making. The most important aspect for me is the influence of Stokes number on particle velocity through the probe sample volume, this is nice. That said, I think that there are a couple of general and more specific issues that I would prefer to see addressed before publication.

The flow model includes the airframe, or possibly a section of wing and the canister plus pylon, and a hemispherical dome approximation of the probe. Some more details about how much of the aircraft is included in the model would be useful and if the flows

at each of the four hard points are the same or different. No specifics are given for modelling the DC-8, it is unclear whether the flows at the probe location on the DC-8 shall be similar or the same as those on the Falcon. This is important as one of the conclusions relates to minimising the difference between TAS and PAS by judicious positioning of the probe on an aircraft.

The angle of attack (AOA) of the Falcon is briefly mentioned in section 2.1.2. It appears that the AOA is held constant for all flight conditions used in the simulations such that the orientation of the probe is always aligned with the local flow, or at least insofar as the error associated with the misalignment is negligible. Given that the simulations are done over a very large range of pressure and TAS and that an aircraft's AOA may be expected to change by degrees over this range, it would be useful to see some justification for this statement. No mention is made about these matters with regards to the DC-8 either.

This work appears to approximate the probes with a semi-hemispherical dome. It would be a significant addition (and an unreasonable request) to include the actual probe geometries. However a comment comparing this work to that of Weigel et al. (2016) and Korolev, Emery, and Creelman (2013), both already cited and which use accurate probe models, would be very interesting given the influence of the probe design on the local flow distortions and the apparent similarities in PAS/TAS ratios seen.

**2  Specific Comments**

More specific comments are included here. I have included the page.line number/s to assist although some of these comments may apply to multiple places in the text.

3.31 The manuscript uses 'airspeed' extensively throughout. It would be useful in the context of airborne measurements to clarify that true air speed is used as

opposed to any of the other definitions used with regards to aircraft. TAS and PAS are defined later at 5.20. 'Speed' is mentioned in section 2.1 but this clarification may be introduced earlier if more convenient/coherent. See comment 7.22 below for more.

4.3 Air speed uncertainties are more relevant here than those in static pressure, can these be related to air speed errors for the range of conditions relevant to this work?

4.5 Are the uncertainties of the DC-8 MMS known? Are they comparable with those of the CMET?

4.30 In my opinion it is inaccurate to say that most probes have pitot tubes close to the sample volume. The DMT PIP- or CIP-based instruments do, the AIMMS does (although is less relevant as it does not measure all that close to particle instrument sample volumes), but (most/all?) other instruments do not. So actually the reverse of what is stated is the case. I'd suggest adding the importance of local PAS measurements to the conclusion of the manuscript with discussion about when it is most/least necessary based on what is being measured.

5.26 Positional errors are discussed throughout the manuscript and I found the usage confusing. Positional error, when defined in section 2.1.2, relates (as I read it) to the difference between TAS and PAS where TAS is the free stream air speed and PAS is the airspeed in the sample volume in the probe. However in 9.9 it seems that positional error possibly also refers to the difference between the air speed measured at the pitot on the probe and that at the sample volume of the probe (due to the difference in PAS measured by the long and short pitot tubes). However, there is no specific mention/quantification of the difference in air speed between the pitot and sample volume. Please clarify the usage if possible.

6.16 Does "simplified three-dimensional model" mean a hemispherical dome as shown

in Fig 4? If so this is quite the simplification for a CAPS for example. So mention of expected uncertainties due to this simplification would be useful.

7.22 The use of 'airspeed' throughout is not as clear as it should be. In this sentence it is used to describe the velocity of a particle relative to the volume of air that contains it (slip velocity). Previously it has been used to describe the velocity of the aircraft and/or probe relative to the air. The difference is very important in this work so I suggest defining unique terms for consistent use throughout. TAS and PAS are traditionally used for the latter case so I'd suggest the use of slip velocity or some other term for the former.

8.23 Mention should be made to confirm if aircraft angle of attack changes with TAS in this model (here and elsewhere).

13.11 There is something missing in the description of the Vargas experimental set up which made interpretation difficult. It turned out that this is the fact that the droplet is 'dropped' perpendicular to the plane of the rotating aerofoil and therefore has a velocity in that plane of zero when far from the aerofoil. An extra sentence to more fully describe the experiment that you are simulating would be useful. Also suggest using slip velocity instead of $U_{rel}$ as this is a commonly known term. How is 'breakup' that occurs around 60 ms$^{-1}$ defined/identified given that the images in figure 14 contain only a single particle?

14.10 It is not obvious to me why the varying slip velocity case is different from the invariant one. An explanation would be appreciated.

17.4 VOF has not been defined previously.

17.5 Has there been discussion about "wiggling" behaviour previously? More specific language is required here.

17.6 Have Taylor instabilities been discussed previously?

**3   Technical Corrections**

There are quite a few basic notation inconsistencies, spelling mistakes, and technical errors that make following the text difficult and detract from the quality of the manuscript. Below are listed the technical corrections that I have caught, given as page.line numbers.

3.32 "per default" is quite an odd expression. "By default" would be more typical although in this instance I think the sentence makes sense without either, ie "The DLR Falcon is equipped with a..."

4.3 Does "pilot-induced manoeuvres" mean turns but not fluctuations due to autopilot corrections or turbulence? Clarification of this phrase would be helpful.

4.19 Is "composed instrument" supposed to read "compound instrument"?

4.21 A OAP does not measure diameter directly, the user/software determines the size from the image recorded by the array. Different size metrics are defined in section 3.2.2 and so could be referred to here.

4.23 I'm not sure what is meant by "named shutter speed of the camera".

4.24 The SID is a scattering instrument not an OAP so should be removed from this list.

5.19 It seems as if "attack angle" here is used in reference to the alignment of the probe to the local air flow. This is confusing as "angle of attack" is traditionally used in reference to the alignment of the aircraft to the air flow (and indeed appears to be used in this way in 5.16). This should be clarified.

5.21 I'd prefer that "measured" is removed as the PAS is the airspeed at the probe location whether it is measured or not.

6.16 Change PMS and add canister to sentence to read "...Particle Measurement Systems canister..."

6.21 Do you have a reference for snappyhexmesh?

6.24 There is some confusing notation regarding $U$, $\boldsymbol{U}$, and U. The usage should be consistent, particularly between the italic and upright versions. One can reasonably assume that $U_0$ is defined along the free stream direction given $|\boldsymbol{U}_0| = \text{TAS}$ but an explicit definition of the axes along with consistent nomenclature would make this is clearer.

7.13 Change "as" to "an".

7.16 Change "model including..." to "model to include...".

7.17 Change "not fully agrees..." to "does not fully agree...".

7.27 I'm not sure what is meant by "...both in the spatial and temporal discretization". Should "in" be replaced by "with"?

8.26 This sentence uses upright U, should they be italic?

9.3 The sentence starting "The simulation results refer to..." is unclear. Is this "The figure refers to..." or "The simulation results are relative to..." or something else?

9.9 See discussion on positional errors in the previous section but in this sentence it would be useful to know why and how the longer pitot tube effects to positional error. This shall assist in understanding the uncertainties when applying the methods discussed to instruments with different pitot tubes or even using pitot tubes mounted near a probe but not as part of the instrument.

9.17 Typo in "However".

10.9 The test case u100_p900 is not actually included in Figure 8.

11.16 Specify which air speed you are referring to here.

12.18 It may be pedantic but Figure 12 shows circular images (which admittedly one assumes are of spherical drops).

12.23 The $x$ and $y$ dimensions should be defined in terms the array and with time (in terms of which the image is described in 12.16) so that the reader understands the aspect ratio values. This is clarified in 13.1 but should be brought forward to this point.

12.24 The markers in Figure 13 are not black.

14.3 The symbol for Weber number should be italics so that it is more obvious in text.

15.1 Add what threshold value you have used in your greyscale images when finding image area.

18.22 This is cited (incorrectly) in the body as Osborne and Cotton. The full author list is required here.

45.6 The data DOI needs updating.

Comments on figures:

Fig 2 Many of the figures are shaded by number of seconds, for clarity it would be useful to add "...number of seconds of data at ?? Hz."

Fig 4 The free stream flow should be marked on this plot. There is some confusion regarding the orientation of the flows in the model between figures 4, 5, and 7. In figure 4, if the free stream is oriented with the figure then it appears as if the angle

of attack (AOA) of the aircraft is not accounted for and the probe is not aligned with the flow. In figure 5, the free stream is aligned with the probe and so the AOA has been accounted for, however for the given flight conditions I assume that the AOA is changing and this is not mentioned (see previous comments regarding AOA). In figure 7, the free stream seems to be coming up to the probe from below (which is opposite to the case shown in figure 4) or perhaps aligned with the probe, it is difficult to tell.

Fig 5 The pitot tube seems to be located diagonally off centre from the dome. Does the schematic represent where the pressure/velocity was measured or was it measured in front of the stagnation point of the dome (I assume that this is where the sample volume is)?

Fig 7 It would be useful to use the same colour scale parameters in this figure as in figure 4. When I converted from ratio to difference the values appear inconsistent between these two figures, is this an issue with my maths/understanding of what is being presented or an actual inconsistency between the two plots?

Fig 13 The main text and caption refer to black markers, maybe it's my display/print but there doesn't appear to be any black markers. The A-LIFE and ATom-2 data have similar shades of grey, completely different colours would be significantly clearer. Is there a reason that the red trace, the mean for the ATom-1 data does not approach unity for small particles?

Fig 16 There appears to be multiple data points for the same TAS values (most evident for 125 ms$^{-1}$), is this a plotting error or does it illustrate the different test cases? If the latter it should be mentioned that P and T don't make a lot of difference.

---

## Short Comment (SC1) · 2 May 2019

On behalf of all authors of Weigel et al. (2016) I would like to address some issues in the manuscript concerning the statements about our work.

1. Statement in your manuscript (page 2, lines 27-30)
   "Recently Weigel et al. (2016) proposed a more general correction method for compressible flow mainly based on thermodynamical calculations. However, this empirical approach is only partially considering the size-dependent effect of particle inertia on the detected concentration. Furthermore, flow disturbances induced by the aircraft wings are not considered by Weigel et al. (2016)."

   The definition of $\xi$ is not empirical but exclusively based on thermodynamic con-

siderations and is indeed essential to account for the compressed sample air volume under measurement conditions compared to ambient (undisturbed) conditions. The inclusion of $\mu$ in the overall correction provided in Weigel et al. (2016) factually considers (not 'only partly') the size dependent effect of particle inertia. Flow disturbances by the aircraft wings are not explicitly resolved as they were not observable as such and may be implied in the compressed condition under which the measurement occurs. So, one could understand your chosen formulation as misleadingly pejorative.

Please change the phrasing in the manuscript to account for this issue.

2. Statement in your manuscript (page 11, lines 20-22)
"Weigel et al. (2016) provides a more rough estimation based on the concept that the air compressibility effect will cause particle accumulation near the instrument. However, the concentration at the wing instrument is apparently larger only because particles are slowed down and stay longer in the corresponding region (see Fig. 9)"

The content of the referred paper might not be fully understood by readers of your paper, since the concept is not that particles accumulate. In Weigel et al. 2016, it is explicitly stated: "Under the preliminary assumptions that the particle number per mass $M$ of the air sample is not affected by compression (i.e. remains constant and thus $\frac{n_{amb}}{M} = \frac{n_{meas}}{M}$)..."

This is the case if particles and air volume get compressed equivalently. Your added statement, however:

"the concentration at the wing instrument is apparently larger only because particles are slowed down and stay longer in the corresponding region"

is one of the messages provided by Weigel et al. (2016) from the contrary perspective: small particles are capable to move out of their initial (undisturbed) state, induced by the compression region upstream of the underwing probe, due

to the approaching aircraft. Larger particles are less capable to get moved out of this undisturbed state due to their inertia.

Please make the relation to Weigel et al. (2016) better visible.

**Reference**

Weigel, R., P. Spichtinger, C. Mahnke, M. Klingebiel, A. Afchine, A. Petzold, M. Krämer, A. Costa, S. Molleker, P. Reutter, M. Szakall, M. Port, L. Grulich, T. Jurkat, A. Minikin, and S. Borrmann, 2016: Thermodynamic correction of particle concentrations measured by underwing probes on fast flying aircraft. Atmos. Meas. Tech., 9, 5135-5162. doi: 10.5194/amt-9-5135-2016

---

## Referee Comment (RC2) · Anonymous Referee #2 · 6 Jun 2019

General comments

The paper takes simulations of the airflow around the wing and canister mount of the DLR Falcon aircraft (the canister mount being the location of the aircraft's cloud microphysics instruments) and uses these to determine biases and corrections upon cloud microphysics measurements. It also examines how this airflow can induce droplet deformation and breakup. A particular aspect of this work is its use of compressible flow in the simulations which allow application to airspeeds relevant to the faster speeds flown by the Falcon.

In general the work is entirely relevant to AMT and very worthwhile. Significant biases can exist in aircraft cloud microphysics measurements and this addresses one of those sources. In particular the work regarding biases in concentration due to airflow are

incredibly useful for the community. There are two overarching limitations though, that I feel the authors should address, that I will mention here in the general comments. I will add more detail in the specific comments section.

The first of these limitations is that the simulations do not include the aircraft fuselage. There are three basic elements that could distort the flow along a particle trajectory before measurement. These are the instrument itself (represented in this work by a general instrument canister), the aircraft wing and the aircraft fuselage. The authors have chosen to include two of these three items in the simulations. They need to justify why the instrument and the wing are important, but the fuselage is not. I am not claiming that the fuselage is important, as I cannot say for certain that it is or it is not, but the authors must justify its exclusion.

The second of the limitations is the discussion around image deformation and drop deformation and breakup. Figure 16, showing the breakup diameter for various airspeeds is a very useful plot as it shows the limits of our measurements. However the discussion of droplet deformation seems to be less rigorous than the work regarding flow induced biases and given the other mechanisms which may induce IMAGE distortion, it is difficult to be certain that the distorted images are definitely the result of distorted droplets in the manner described here. Some of the theory, observations and discussions do not seem consistent. I would suggest one of two options. Either this section needs rewriting, being much more careful about consistency and ensuring it is very clear what is discussion and what is conclusion, or alternatively this section could be significantly reduced in size or removed – the section discussing efficiencies is interesting enough to stand alone. As a final general recommendation, although the English in the paper is mostly very good (and much better than my foreign language skill would permit me to write in my non-native tongue), there are times when it is obvious that the paper is written by a non-native English speaker and the wording is difficult to understand. I would therefore recommend that the authors have the final version proof read by a native English speaker.

Overall, the paper covers important aspects of the cloud microphysics measurement system and I absolutely recommend publication with the changes outlined in this review.

Specific Comments

P2L14 Explain why the temperature and pressure further affect the aerosol and cloud measurements – what mechanism are you referring to.

P2L18. Are droplets appearing deformed because they are deformed due to aerodynamic forces or are they appearing deformed because of a bias with the instrument system (off axis flow, incorrectly measured particle speed).

P2L32 compressible flow is not a hypothesis

P3L2 and throughout aircraft is probably a better word than plane or airplane in technical writing.

P4L24 SID-2 is an open path OPC, not an OAP. Note also that SID-3, despite taking images, is not and OAP either as it uses a 2D CCD, not a line of photodiodes.

P5L15 Insert the word measured before "dynamic pressure"

P5L23 Is the overestimation of airspeed an overestimation of PAS or TAS. Be specific.

P6L4 Have you checked the conservation of total pressure at the two pitot tubes? This would give a good check that their calibrations are not introducing bias.

P6L19 The authors have felt it necessary to make the edge length 10 times the canister length to avoid biases from the boundary. Although I appreciate that this was no-doubt chosen because it was a round number, that was at least as large (but probably larger) than necessary, I would guess that within this domain one may find either the aircraft fuselage itself or air which had been modified by flow around the fuselage. It may well be that this is not the case, or that the flow distortion caused by the fuselage does not impact the particle paths, but the authors should show that this is the case or they

should provide some limits on the potential effects of the fuselage. They should also describe the effect of the wing itself, so that it is clear to the reader that both the wing and the instrument canister have an impact upon the flow. This will certainly feed into decisions made by future studies into flow effects on microphysics measurements.

P8L14 It would be nice to highlight the test case on all plots, either by circling the data point or putting the line in bold or some other method.

P8L21 replace probe and free stream with local and free stream

P8L28 it would be good to see the incompressible solution on the charts for comparison.

P9L1 At first reading I thought the 1% error in ps was causing a 23% error in U. It may be worth rewriting this sentence to ensure other readers don't make the same mistake.

P9L2 How much do we really care about temperature? Later in the paper the authors refer to the fact that there is a compression of air and a bias proportional to density which is in turn proportional to temperature. However, the temperature increase is a maximum of 3%. The authors also mention that the temperature is measured "round the back" of the probe so may not be that relevant to the sample volume anyway. The authors should simply consider if the bias caused by temperature increase is worth considering, given the other uncertainties, and if it is they should explain at this point why it is of interest.

P9L11 "Well represent" is an ambiguous term. The points from the simulations on figure 6c do not show the upside down U shape that the measurements show. The authors should state some measure of the actual discrepancy and why they feel that this is sufficient.

P9L14 What does "installed at the back" mean? Give some better description of the location of the temperature sensor and if it is not close to the sample volume then describe the expected error due to the position and whether this is sufficient – see

above comment re temperature in general.

P9L31 In what way is the stokes number modified and why?

P9l33 Equation(4) which velocity is used here for U? See later comments regarding stokes number.

P10 It is very impressive how well the data fit on the sigmoid curves on fig 8. This clearly forms an excellent correction factor. However a few items may aid the reader in understanding the analysis that is occurring here. Firstly it would be good to indicate that the fit lines used are from equation (6). Secondly it should be noted that $\alpha$ is of the order 1 and contributes negligibly to the parameters x0 and k0, so should probably be removed. Thirdly the authors should note that eˆ(-k0(log(stk)-x0) can be rearranged to b*stkˆa which is a much simpler form. Also if they remove the dependence of k0 and x0 upon $\alpha$, then a and b simply become fitting constants. In fact b=10ˆ(x0*log(eˆk0)) and a=-log(eˆk0). Which makes b approximately -1. Fourthly it may be useful for the authors to rearrange the form slightly as this may help the points all converge onto 1 line. They should note the classic inlet efficiency equation from Belyaev & Levin doi:10.1016/0021-8502(74)90130-X, equation (5) Efficiency = 1+(u0/u-1)*sigmoid_function_of_stk_only Where u0 is the far field velocity and u is the velocity at the inlet. This basically states that the deviation from unity efficiency is proportional to u0/u. This form may be useful to the authors, perhaps with u0/u replaced by $\alpha$. Fifthly, as described previously, the authors should consider whether it is really worth including temperature in this analysis as it is a relatively small effect and temperature may not be well measured by the probe.

P11L14 Do the authors actually mean mobility in the sense that it is used in aerosol science, e.g. for a scanning mobility particle sizer instrument? If not then change this word.

P11L26 The word positional error seems like an odd choice. Perhaps the authors mean distortion errors or something similar.

P11L27 did the authors mean 100 um? 10 um particles seem significantly affected by the flow.

P11L31 rephrase. I think you want to say something like we adjust the calibration constants in the data logging software so that the pitot tube reports an air speed close to TAS rather than PAS. When this is reported to the instrument it causes the (insert info about lines imaged at the correct rate), but the PAS may still be recovered later by using the recorded pressures and the correct calibration constants.

P12L14 The images shown in figure 12 are clearly distorted. However, it is not at all clear to me that this is a distortion of the droplets or just the images. In particular the images show a skew. This skew could be due to misalignment of the particle trajectory with the instrument axis, or perhaps it could be due to shear flow distorting the particles. This skew in itself causes an apparent lengthening of the particles in the x direction – you will note that the farthest shadowed pixels on a droplet do not fall on the same line. It is not clear if any corrections for this have been made. Until a model can account for the skew distortion and the flattening I think it is difficult to claim which mechanism is responsible. I think it is appropriate here to discuss how flattened the droplets are expected to be and to suggest it is a possible mechanism in the context of other possible mechanisms. But I don't feel it has been proven here that droplet flattening is the definite cause. Indeed the authors have the flow data to answer the question – are the particles travelling parallel to the probe axis?

P13L5 It appears that airspeed errors of 10-20% would be perfectly sufficient to centre the distribution of dy/dx over 1.0 for Atom-1. The a-life and Atom-2 data appear to show the opposite effect to that which would be expected from the droplet distortions described here. Please explain.

P13L21 Be specific about what a small deviation is. Give the amount.

P14L23 There is no evidence for Taylor instabilities in Fig 13. Visually the scatter in the data looks to be entirely consistent with the error bars that the authors have put on

the data. The authors would need to do some further analysis of the data scatter and if they find that the scatter is too large to be explained by the uncertainties, then and only then, should they invoke a mechanism to explain the extra scatter.

P14L28 Something here is not really matching up. The authors state that for a 200 um droplet they have We of 2.5 and it cannot be considered spherical, however close to that diameter on figure 13 the particles seem to generally have dy/dx within one error bar of unity. The author appears to be suggesting that observed broken up particles are caused by the mechanism here, or at least that some aircraft can go fast enough that droplets will be broken up so cannot be observed by the probes above a certain diameter. But the points in Figure 15 show droplets going up to aspect ratios of 1.5 to 8. The only times I have seen such distorted particles has been when the refresh rate of the diode array has been clearly wrong. It may be that the authors have extrapolated the data for figure 15 past the break up point or perhaps there is something I am not able to piece together. But I can only suggest that section 3.2 needs a really thorough rewrite in order to ensure the arguments being made are consistent with the data and the model and to be clear what is a discussion and what is a conclusion.

P14L29 I feel that the items discussed in section 3.2.2 are probably an overanalysis of the data. Again the authors should be clear about what is a conclusion and what is a discussion and before they begin suggesting corrections for an effect they first need to be clear that they have shown the effect is real.

P15L25 should be clear that size is diameter rather than radius

P15L30 see comment above re corrections

P16L5 This set of steps would be much simplified to the point of being obvious if the advice above is taken regarding making the equations a bit clearer and specifying which air speed is being used for Stk.

P17L3 Again it is not clear to me that droplet deformation as a cause of the image

distortion has been proven.

P17L13 Stating this as a potential limit for the drop size that can be seen as a function of aircraft speed is I think very interesting, but my gut feeling is that observed droplet breakups are caused by impacts or near impacts with the probe.

P26Fig2 It would be nice to highlight the test case.

P27Fig3 The lines of best fit are linear – the word polynomial should be saved for higher order equations. Replace with Linear. All plots should at least have x and y axes that start at the same value. I would ideally like to see the same max value on the x and y axes too, but I appreciate that for Fig2d, this may not be appropriate.

P27Fig4 ensure that 0 is marked on the colour bar.

P28Fig5 It would be nice to see the test case highlighted

P29Fig6 It would be nice to see the test case highlighted

P36Fig13 The lines have no proper description of what they mean – are they averages over some size range? I think they are not useful anyway and should probably be removed unless the authors can argue a good reason for them. Also 2 different shades of grey should not be used. It is hard to tell them apart.

---

## Author Comment (AC1) · 16 Dec 2019

We wish to thank the reviewer for carefully reading our manuscript and for providing suggestions which helped us to improve the manuscript. In the following, the questions and comments raised by the reviewer are marked in blue and our answers are written in black.

This manuscript is a useful addition to the growing body of work investigating the influence of airborne particle instruments on the measurements that they are making. The most important aspect for me is the influence of Stokes number on particle velocity through the probe sample volume, this is nice. That said, I think that there are a couple of general and more specific issues that I would prefer to see addressed before publication.

The flow model includes the airframe, or possibly a section of wing and the canister plus pylon, and a hemispherical dome approximation of the probe. Some more details about how much of the aircraft is included in the model would be useful and if the flows at each of the four hard points are the same or different. No specifics are given for modelling the DC-8, it is unclear whether the flows at the probe location on the DC-8 shall be similar or the same as those on the Falcon. This is important as one of the conclusions relates to minimizing the difference between TAS and PAS by judicious positioning of the probe on an aircraft.

Related to this question, we give more detail below. Generally, the flow under the wing is slightly varying along the wing since the wing cross section varies between the aircraft fuselage and the wing tip. Because of this, small differences between the hard points are possible. We haven't modelled the DC-8 since our results show that the concentration depends only on the ratio between the local air flow and the free stream air flow.

The angle of attack (AOA) of the Falcon is briefly mentioned in section 2.1.2. It appears that the AOA is held constant for all flight conditions used in the simulations such that the orientation of the probe is always aligned with the local flow, or at least in so far as the error associated with the misalignment is negligible. Given that the simulations are done over a very large range of pressure and TAS and that an aircraft's AOA maybe expected to change by degrees over this range, it would be useful to see some justification for this statement.

We have performed numerical sensitivity studies related to the dependency of the results on AOA. Figure 1 (in the reply) shows the deviation between PAS and TAS as a function of ambient pressure color-coded with the angle of attack (AOA) for the arbitrarily chosen TAS range from 152.5 m/s to 157.5 m/s from measurements during the SALTRACE campaign. A TAS range had to be chosen for this figure because the deviation between PAS and TAS depends also on TAS. However, the picture is similar for other TAS ranges and shows that the AOA has only a minor impact on the measurements ((PAS-TAS)/TAS changes less than 2% when the AOA is changed). Therefore, we chose a median AOA value (4°) for our analysis.

[Figure]

*Figure 1. Deviation between PAS and TAS as a function of ambient pressure color-coded with the angle of attack (AOA). Markers represent measurements at 1Hz collected during the SALTRACE campaign.*

No mention is made about these matters with regards to the DC-8 either.

On the DC-8, CAPS is mounted near the wing tip, i.e. it is less affected by the AOA, since the wing chord and the wing thickness is smaller at this part of the wing.

This work appears to approximate the probes with a semi-hemispherical dome. It would be a significant addition (and an unreasonable request) to include the actual probe geometries. However, a comment comparing this work to that of Weigel et al. (2016) and Korolev, Emery, and Creelman (2013), both already cited and which use accurate probe models, would be very interesting given the influence of the probe design on the local flow distortions and the apparent similarities in PAS/TAS ratios seen.

Figure 6b and 6c in the manuscript show the measured and simulated differences between p and PAS at the probe location and the corresponding values at free stream condition. The simulated values are well within the measured range. Therefore, we think that the simplified instrument geometry is a valid assumption.

Specific Comments

More specific comments are included here. I have included the page.line number/s to assist although some of these comments may apply to multiple places in the text.

**3.31** The manuscript uses 'airspeed' extensively throughout. It would be useful in the context of airborne measurements to clarify that true air speed is used as opposed to any of the other definitions used with regards to aircraft. TAS and PAS are defined later at 5.20. 'Speed' is mentioned in section 2.1 but this clarification may be introduced earlier if more convenient/coherent. See comment 7.22 below for more.

We changed this in the manuscript as suggested.

**4.3** Air speed uncertainties are more relevant here than those in static pressure, can these be related to air speed errors for the range of conditions relevant to this work?

Yes, if we assume the errors are only affecting the static pressure. In this case, a 1% overestimation of the static pressure will result in a 0.5% overestimation of the air speed according to the Bernoulli equation.

**4.5** Are the uncertainties of the DC-8 MMS known? Are they comparable with those of the CMET?

Yes, the two uncertainties are similar and represent errors smaller than 2% on the total pressure. Figure 2 shows the deviation of the total pressure between CAS and the Falcon CMET system during SALTRACE and between CAS and the DC-8 MMS system during ATom-1.

[Figure]

*Figure 2. Deviations of total pressure measured by CAS and by the Falcon CMET system during SALTRACE (left), and the DC-8 MMS system during ATom-1 (right), respectively.*

**4.30** In my opinion it is inaccurate to say that most probes have pitot tubes close to the sample volume. The DMT PIP- or CIP-based instruments do, the AIMMS does (although is less relevant as it does not measure all that close to particle instrument sample volumes), but (most/all?) other instruments do not. So actually, the reverse of what is stated is the case. I'd suggest adding the importance of local PAS measurements to the conclusion of the manuscript with discussion about when it is most/least necessary based on what is being measured.

We changed the text in the paper as suggested. It reads now:

"As we described, OPC and OAP measurements depend on the flow, therefore wing-mounted instruments are sometimes equipped with flow sensors to constrain local conditions."

In addition, we added text to the conclusion of the manuscript explaining the importance of local PAS measurements.

5.26 Positional errors are discussed throughout the manuscript and I found the usage confusing. Positional error, when defined in section 2.1.2, relates (as I read it) to the difference between TAS and PAS where TAS is the free stream air speed and PAS is the airspeed in the sample volume in the probe. However, in 9.9 it seems that positional error possibly also refers to the difference between the airspeed measured at the pitot on the probe and that at the sample volume of the probe (due to the difference in PAS measured by the long and short pitot tubes). However, there is no specific mention/quantification of the difference in air speed between the pitot and sample volume. Please clarify the usage if possible.

With the term "position error", we refer to the difference between TAS and PAS. For this reason, a longer pitot tube will reduce the position error because the difference of the pressure at the probe and at free stream is exponentially decreasing as a function of the distance from the probe head. We assume the pitot tube measurements to be representative for the sampling area (see Fig. 5 and text of the revised manuscript). Thus, using a longer pitot tube, implies the sampling area also to be at a larger distance from the main instrument body. This is the case for CAPS.

6.16 Does "simplified three-dimensional model" mean a hemispherical dome as shown in Fig 4? If so this is quite the simplification for a CAPS for example. So mention of expected uncertainties due to this simplification would be useful.

As mentioned previously, Figure 6 in the manuscript shows the measured and simulated differences in pressure and air speed at the probe in comparison with free stream conditions. The simulated values are well within the measured range. Therefore, we think that the simplified instrument geometry is a valid assumption.

7.22 The use of 'airspeed' throughout is not as clear as it should be. In this sentence it is used to describe the velocity of a particle relative to the volume of air that contains it (slip velocity). Previously it has been used to describe the velocity of the aircraft and/or probe relative to the air. The difference is very important in this work so I suggest defining unique terms for consistent use throughout. TAS and PAS are traditionally used for the latter case so I'd suggest the use of slip velocity or some other term for the former.

We modified the text and the figures, according to your comment and now use "slip velocity".

8.23 Mention should be made to confirm if aircraft angle of attack changes with TAS in this model (here and elsewhere).

The simulations presented in the manuscript were made for a fixed AOA (4°). This AOA was selected to represent the median conditions as visible in Figure 3. Also, it can be seen from Figure 1 that changes of the AOA have only a minor impact on the relative deviation of PAS from TAS.

[Figure]

*Figure 3. Frequency of AOA vs. true air speed (TAS) during the SALTRACE campaign in 2013.*

13.11 There is something missing in the description of the Vargas experimental set up which made interpretation difficult. It turned out that this is the fact that the droplet is 'dropped' perpendicular to the plane of the rotating airfoil and therefore has a velocity in that plane of zero when far from the airfoil. An extra sentence to more fully describe the experiment that you are simulating would be useful. Also suggest using slip velocity instead of Urel as this is a commonly known term.

We extended the description of the Vargas experiment to clarify it. We also substituted Urel with Uslip in the entire manuscript.

How is 'breakup' that occurs around 60 ms−1 defined/identified given that the images in figure 14 contain only a single particle?

According to Vargas (2012), droplet breakup is occurring at the edge of the droplet by a stripping mechanism. In the volume of fluid (VOF) simulations the breakup is identified by considering the number of regions in the simulation which contain a liquid phase. If more than one such region is identified, the droplet is counted as "broken".

14.10 It is not obvious to me why the varying slip velocity case is different from the invariant one. An explanation would be appreciated.

According to Vargas (2012), the droplet in case of a varying slip velocity has more time to adjust to the flow. We extended the description and added a reference to Vargas (2012).

Corrected.

17.5 Has there been discussion about "wiggling" behavior previously? More specific language is required here.

According to the comment, we removed the term "wiggling" and reformulated the sentence.

17.6 Have Taylor instabilities been discussed previously?

Yes, they were mentioned on 14.22. We reformulated the sentence and removed this term from the conclusions.

**3 Technical Corrections**

There are quite a few basic notation inconsistencies, spelling mistakes, and technical errors that make following the text difficult and detract from the quality of the manuscript. Below are listed the technical corrections that I have caught, given as page. line numbers.

3.32 "per default" is quite an odd expression. "By default" would be more typical although in this instance I think the sentence makes sense without either, ie "The DLR Falcon is equipped with a..."

Done.

4.3 Does "pilot-induced manoeuvres" mean turns but not fluctuations due to auto-pilot corrections or turbulence? Clarification of this phrase would be helpful.

Yes, according to the cited article it refers only to pilot-induced maneuvers. Auto-pilot corrections or 'normal' turbulence probably result in even smaller errors.

4.19 Is "composed instrument" supposed to read "compound instrument"?

We modified the text and now leave out "composed".

4.21 A OAP does not measure diameter directly, the user/software determines the size from the image recorded by the array. Different size metrics are defined in section 3.2.2 and so could be referred to here.

We modified the sentence including a reference to 3.2.2.

4.23 I'm not sure what is meant by "named shutter-speed of the camera".

We modified the sentence and now leave out "named shutter-speed of the camera".

4.24 The SID is a scattering instrument not an OAP so should be removed from this list.

SID has been removed from the list of OAP.

5.19 It seems as if "attack angle" here is used in reference to the alignment of the probe to the local air flow. This is confusing as "angle of attack" is traditionally used in reference to the alignment of the aircraft to the air flow (and indeed appears to be used in this way in 5.16). This should be clarified.

Your comment is correct. To avoid misunderstanding, we modified the sentence.

5.21 I'd prefer that "measured" is removed as the PAS is the airspeed at the probe location whether it is measured or not.

Done.

6.16 Change PMS and add canister to sentence to read "...Particle Measurement Systems canister..."

Canister in mentioned later in the same sentence so that we think it is not necessary to add it here.

6.21 Do you have a reference for snappyhexmesh?

We added a reference to snappyhexmesh:

Montorfano, A (2017) Mesh generation for HPC problems: the potential of SnappyHexMesh, doi:10.13140/RG.2.2.25007.53923

6.24 There is some confusing notation regarding $U$, $\textbf{U}$, and U. The usage should be consistent, particularly between the italic and upright versions. One can reasonably assume that U0 is defined along the free stream direction given|U0|=TAS but an explicit definition of the axes along with consistent nomenclature would make this is clearer.

Done. For clarity, we included a new table in the revised manuscript giving an overview of the different velocities and speeds.

7.13 Change "as" to "an".

"As" was correct, but "an" was missing. Now it reads: "Droplet breakup, as an effect of the instability …"

7.16 Change "model including…" to "model to include…".

Done.

7.17 Change "not fully agrees..." to "does not fully agree...".

Done.

 I'm not sure what is meant by "...both in the spatial and temporal discretization". Should "in" be replaced by "with"?

We think that "in" is correct, but modified the sentence slightly to make it more clear. It now reads as: "... both in the spatial and in the temporal discretization ..."

8.26 This sentence uses upright U, should they be italic?

Corrected.

9.3 The sentence starting "The simulation results refer to..." is unclear. Is this "The figure refers to..." or "The simulation results are relative to..." or something else?

Improved. The sentence reads now "The simulation results in Fig. 6 are valid ..."

9.9 See discussion on positional errors in the previous section but in this sentence it would be useful to know why and how the longer pitot tube effects to positional error. This shall assist in understanding the uncertainties when applying the methods discussed to instruments with different pitot tubes or even using pitot tubes mounted near a probe but not as part of the instrument.

See also our reply to comment 5.26 (p. 4). The text in the manuscript has be revised.

9.17 Typo in "However".

Although there is the term "howsoever", we now use "nevertheless".

10.9 The test case u100_p900 is not actually included in Figure 8.

The case u100_p900 has been added to Fig. 8.

11.16 Specify which air speed you are referring to here.

As recommended, we use "probe air speed" to make the message more clear.

12.18 It may be pedantic but Figure 12 shows circular images (which admittedly one assumes are of spherical drops).

We changed the text as follows: "Most of the droplet images are circular…"

12.23 The x and y dimensions should be defined in terms the array and with time (in terms of which the image is described in 12.16) so that the reader understands the aspect ratio values. This is clarified in 13.1 but should be brought forward to this point.

We modified the manuscript as suggested.

 The markers in Figure 13 are not black.

We modified the Figure 13 which now uses three different colors.

14.3 The symbol for Weber number should be italics so that it is more obvious in text.

We modified the Weber symbol in the entire text.

15.1 Add what threshold value you have used in your greyscale images when finding image area.

We have used a threshold value of 0.8. The value is now specified in Sect. 3.2.

18.22 This is cited (incorrectly) in the body as Osborne and Cotton. The full author list is required here.

The SID reference has been removed according to your comment and consequently the reference too.

45.6 The data DOI needs updating.

We are in the process of updating the data DOI. Since we included new data, the update takes more time.

Comments on figures:

Fig 2 Many of the figures are shaded by number of seconds, for clarity it would be useful to add "...number of seconds of data at 1 Hz".

We added the "...number of seconds of data at 1 Hz" in the figure captions where necessary.

Fig 4 The free stream flow should be marked on this plot. There is some confusion regarding the orientation of the flows in the model between figures 4, 5, and 7. In figure 4, if the free stream is oriented with the figure then it appears as if the angle of attack (AOA) of the aircraft is not accounted for and the probe is not aligned with the flow. In figure 5, the free stream is aligned with the probe and so the AOA has been accounted for, however for the given flight conditions I assume that the AOA is changing, and this is not mentioned (see previous comments regarding AOA). In figure 7, the free stream seems to be coming up to the probe from below (which is opposite to the case shown in figure 4) or perhaps aligned with the probe, it is difficult to tell.

To make the message more clear, we changed the color-scale and added an arrow indicating the direction of the free stream flow. We also modified the scale label now using "$(U-U_0)/U_0$" to be consistent with Figure 5.

Fig 5 The pitot tube seems to be located diagonally off centre from the dome. Does the schematic represent where the pressure/velocity was measured or was it measured in front of the stagnation point of the dome (I assume that this is where the sample volume is)?

Correct, the pitot tube is located diagonally off center in the instrument and in the simulations. The schematic represents where the pressure/velocity was measured/simulated. The simulation results refer to the pitot tube static port location.

Fig 7 It would be useful to use the same colour scale parameters in this figure as in figure 4. When I converted from ratio to difference the values appear inconsistent between these two figures, is this an issue with my maths/understanding of what is being presented or an actual inconsistency between the two plots?

We modified the Figure. Now it is showing the density of the air ($(\rho - \rho_0)/\rho_0$). We modified the text accordingly.

Fig 13 The main text and caption refer to black markers, maybe it's my display/print but there doesn't appear to be any black markers.

The A-LIFE and ATom-2 data have similar shades of grey, completely different colours would be significantly clearer.

We modified the figure using now three different colors.

Is there a reason that the red trace, the mean for the ATom-1 data does not approach unity for small particles?

Yes, the bias is introduced by the wrong droplet speed (PAS instead of TAS) for the image reconstruction.

Fig 16 There appears to be multiple data points for the same TAS values (most evident for 125 ms−1), is this a plotting error or does it illustrate the different test cases? If the latter, it should be mentioned that P and T don't make a lot of difference.

For some velocities (TAS) there are more than one test case (see Table 2 in the manuscript for details). We added a clarifying comment in the manuscript.

---

## Author Comment (AC2) · 16 Dec 2019

We thank Peter Spichtinger for his comments on our manuscript. It was not our intention to write pejoratively about Weigel et al. (2016). We modified our manuscript based on his suggestions. In the following, the comments raised by Peter Spichtinger are marked in blue and our answers are written in black.

On behalf of all authors of Weigel et al. (2016) I would like to address some issues in the manuscript concerning the statements about our work.

1. Statement in your manuscript (page 2, lines 27-30)
    "Recently Weigel et al. (2016) proposed a more general correction method for compressible flow mainly based on thermo-dynamical calculations. However, this empirical approach is only partially considering the size-dependent effect of particle inertia on the detected concentration. Furthermore, flow disturbances induced by the aircraft wings are not considered by Weigel et al. (2016)."

The definition of ξ is not empirical but exclusively based on thermodynamic considerations and is indeed essential to account for the compressed sample air volume under measurement conditions compared to ambient (undisturbed) conditions. The inclusion of μ in the overall correction provided in Weigel et al. (2016) factually considers (not 'only partly') the size dependent effect of particle inertia. Flow disturbances by the aircraft wings are not explicitly resolved as they were not observable as such and may be implied in the compressed condition under which the measurement occurs. So, one could understand your chosen formulation as misleadingly pejorative.
Please change the phrasing in the manuscript to account for this issue.

As recommended, we modified the paragraph to make it more clear. The paragraph now reads as follows:

Recently Weigel et al. (2016) proposed a more general correction method for particle concentrations measured by an under-wing instrument. Its first component is a compression correction factor that is based on thermo-dynamical calculations using simultaneous measurements of the instrument's pitot tube. Its second component is a size-dependent correction factor that corrects the effect of the inertia of particles larger than 70 μm, but not for smaller particles.

2. Statement in your manuscript (page 11, lines 20-22)
    "Weigel et al.(2016) provides a rougher estimation based on the concept that the air compressibility effect will cause particle accumulation near the instrument. However, the concentration at the wing instrument is apparently larger only because particles are slowed down and stay longer in the corresponding region (see Fig. 9) "

The content of the referred paper might not be fully understood by readers of your paper, since the concept is not that particles accumulate. In Weigel et al. 2016, it is explicitly stated: "Under the preliminary assumptions that the particle number per mass M of the air sample is not affected by compression (i.e. remains constant and thus n_amb/M= n_meas/M)"

This is the case if particles and air volume get compressed equivalently. Your added statement, however: "the concentration at the wing instrument is apparently larger only because particles are slowed down and stay longer in the corresponding region" is one of the messages provided by Weigel et al. (2016) from the contrary perspective: small particles are capable to move out of their initial (undisturbed) state, induced by the compression region upstream of the underwing probe, due to the approaching aircraft. Larger particles are less capable to get moved out of this undisturbed state due to their inertia. Please make the relation to Weigel et al. (2016) better visible.

The description of the different perspectives of this effect is helpful. However, a discrepancy between the results of Weigel et al. (2016) and our results remains: The inertia correction factor μ of Weigel et al. (2016) is equal to 1.00 for diameters $d_p$ < 70μm, implying that these particles follow the air flow. In contrast, our simulations (see e.g. Fig. 9) show a notable impact of the particle inertia already for a diameter of 10 μm (the μ factor would be around 0.90 for density 1g/cm$^3$) and a strong impact for 50 μm particles (μ around 0.75) with a notable effect on concentration too (see Fig. 10).

We double-checked our results using a simplified numerical approach for the particle movement using our CFD-simulated airflow velocity fields ahead of the instrument as input and accounting for the acceleration due to the drag force on the particles. The drag force was estimated based on Eq. 3.5 of the book "Aerosol Technology: Properties, Behavior, and Measurement of Airborne Particles" by W. C. Hinds. We assumed case *u200_p250* with a particle density of 1 g cm$^{-3}$, a TAS of 200 m s$^{-1}$ and an ambient pressure of 250 hPa. The results of this simplified calculation (see attached Fig. 1) confirm the results presented in our paper and thus imply that inertia needs to be taken into account also for particles with diameters smaller than 70μm. For instance, according to Fig. 1, a 30μm particle still has a velocity of about 179 m/s at the measurement location while the air velocity (PAS) is slowed down to about 142 m/s. Therefore the velocity of a 30μm particle is closer to TAS than to PAS. Even a 5μm particle shows some inertia effects, nonetheless PAS and therefore constant particle number per mass M of the air may be considered a good approximation for small particles. For a 10μm particle, however, the deviation may already be considered significant, because 10 μm particles are about 10% faster than PAS.

The revised paragraph reads as follows:

> Weigel et al. (2016) provide a method that is primarily based on the concept that the air compression near the instrument causes a corresponding densification of the number concentration of airborne particles. Subsequently, they take into account a size-dependent correction factor that corrects the effect of the inertia of large particles. Their inertia correction is mainly assessed on the basis of the circularity of images taken by an OAP at a resolution of 15 μm. Weigel et al. (2016) conclude that particles with diameters $d_p$ < 70 μm follow the airflow and thus require no inertia correction. On the contrary, our simulations (see e.g. Fig. 9) show a notable impact of the particle inertia already for particle diameters of 10 μm (their speed is about 10% faster than the air; particle density 1 g/cm$^3$) and a strong impact for 50 μm particles (about 25% faster than air). These particle simulations are consistent with results (not shown) from a simplified numerical particle motion model using the simulated flow fields (Sect. 3.1.1) as input and Eq. 3.5 of Hinds (1999) (which is based on Clift et al. (1978)) to calculate the drag force on the particles. Therefore, we conclude that inertia needs to be taken into consideration for particles larger than about 5-10 μm.

[Figure]

*Figure 1: Velocities of air and particles of various diameters, relative to the aircraft for TAS=200m/s. The air velocity (thick black line) from the OpenFOAM simulation case u200_p250 of the discussion paper is used as input to calculate the motion of the particles caused by the drag force on the particles. The drag force is calculated based on the slip velocity particle, the corresponding Reynolds number, and Eq. 3.5 of Hinds (1999) which is based on a correlation given by Clift et al. (1978). The horizontal axis is the distance from the approaching instrument head. The grey vertical line marks the approximate location of the particle measurements.*

**References:**

Clift, Grace, Weber: Bubbles, Drops, and Particles; Academic Press, 1978.

Hinds: Aerosol technology: properties, behavior, and measurement of airborne particles; John Wiley & Sons, 1999.

Weigel, R., Spichtinger, P., Mahnke, C., Klingebiel, M., Afchine, A., Petzold, A., Krämer, M., Costa, A., Molleker, S., Reutter, P., Szakáll, M., Port, M., Grulich, L., Jurkat, T., Minikin, A., and Borrmann, S.: Thermodynamic correction of particle concentrations measured by underwing probes on fast-flying aircraft, Atmospheric Measurement Techniques, 9, 5135–5162, https://doi.org/10.5194/amt-9-5135-2016, 2016.

---

## Author Comment (AC3) · 16 Dec 2019

We wish to thank the reviewer for carefully reading our manuscript and for providing suggestions which helped us to improve the manuscript. In the following, the questions and comments raised by the reviewer are marked in blue and our answers are written in black.

The paper takes simulations of the airflow around the wing and canister mount of the DLR Falcon aircraft (the canister mount being the location of the aircraft's cloud micro-physics instruments) and uses these to determine biases and corrections upon cloud microphysics measurements. It also examines how this airflow can induce droplet de-formation and breakup. A particular aspect of this work is its use of compressible flow in the simulations which allow application to airspeeds relevant to the faster speeds flown by the Falcon.

In general the work is entirely relevant to AMT and very worthwhile. Significant biases can exist in aircraft cloud microphysics measurements and this addresses one of those sources. In particular the work regarding biases in concentration due to airflow are incredibly useful for the community. There are two overarching limitations though, that I feel the authors should address, that I will mention here in the general comments. I will add more detail in the specific comments section.

The first of these limitations is that the simulations do not include the aircraft fuselage. There are three basic elements that could distort the flow along a particle trajectory before measurement. These are the instrument itself (represented in this work by a general instrument canister), the aircraft wing and the aircraft fuselage. The authors have chosen to include two of these three items in the simulations. They need to justify why the instrument and the wing are important, but the fuselage is not. I am not claiming that the fuselage is important, as I cannot say for certain that it is or it is not, but the authors must justify its exclusion. The second of the limitations is the discussion around image deformation and drop deformation and breakup. Figure 16, showing the breakup diameter for various airspeeds is a very useful plot as it shows the limits of our measurements. However, the discussion of droplet deformation seems to be less rigorous than the work regarding flow induced biases and given the other mechanisms which may induce IMAGE distortion, it is difficult to be certain that the distorted images are definitely, the result of distorted droplets in the manner described here. Some of the theory, observations and discussions do not seem consistent. I would suggest one of two options. Either this section needs rewriting, being much more careful about consistency and ensuring it is very clear what is discussion and what is conclusion, or alternatively this section could be significantly reduced in size or removed – the section discussing efficiencies is interesting enough to stand alone. As a final general recommendation, although the English in the paper is mostly very good (and much better than my foreign language skill would permit me to write in my non-native tongue), there are times when it is obvious that the paper is written by a non-native English speaker and the wording is difficult to understand. I would therefore recommend that the authors have the final version proof read by a native English speaker. Overall, the paper covers important aspects of the cloud microphysics measurement system and I absolutely recommend publication with the changes outlined in this review.

Specific Comments

P2L14 Explain why the temperature and pressure further affect the aerosol and cloud measurements – what mechanism are you referring to.

The flow around the aircraft modifies pressure and temperature of the air and as a consequence of the gas law also the air density at the instrument's sampling area. Consequently also the particle concentration, in particular of small particles, is affected. Furthermore, the flow distortion also modifies the sampled flow velocity. For example, higher pressure is related to a lower flow velocity (Bernoulli equation, energy conservation).

We have updated the manuscript.

P2L18. Are droplets appearing deformed because they are deformed due to aerodynamic forces or are they appearing deformed because of a bias with the instrument system (off axis flow, incorrectly measured particle speed).

They can appear deformed because of both mechanisms:

1.) Droplets may appear as deformed on the images, but they are not deformed on reality. This is the case if the camera does not use the correct velocity for taking the images.

2.) Large droplets are deformed due to aerodynamic forces. Both citations refer indeed to shape distortion mechanism at the droplet surface.

We rephrased the paragraph to make it more clear.

We reworded the text and skipped the word 'hypothesis'.

P3L2 and throughout aircraft is probably a better word than plane or airplane in technical writing.

Done. We now only use the word aircraft.

P4L24 SID-2 is an open path OPC, not an OAP. Note also that SID-3, despite taking images, is not and OAP either as it uses a 2D CCD, not a line of photodiodes.

We removed the SID from the list of instruments.

P5L15 Insert the word measured before "dynamic pressure"

Done.

P5L23 Is the overestimation of airspeed an overestimation of PAS or TAS. Be specific.

This statement is generally valid, i.e. to any velocity derived from pitot tube measurements using the Bernoulli equation under the assumption of an incompressible fluid. For example, using the Bernoulli formula for incompressible gas together with the nose boom pressure values will lead to an overestimation of TAS. In contrast, the same formula applied to the pitot tube of the wing-probe instrument will lead to an overestimation of the PAS.

We adapted the text.

P6L4 Have you checked the conservation of total pressure at the two pitot tubes? This would give a good check that their calibrations are not introducing bias.

Figure 1 below shows comparisons of the total pressure measured by the pitot tube of the CAPS with measurements of the CMET pitot tube on the DLR Falcon and the MMS pitot tube on the NASA DC-8, respectively. The deviation between the different pressure sensors is less than 2%.

[Figure]

*Figure 1.* Comparison between different total pressure measurements as a function of ambient pressure. Left: Ratio between CAPS pressure measurement and the DLR Falcon CMET pressure measurement. Right: Ratio between the CAPS pressure measurement and the NASA DC-8 MMS pressure measurement.

P6L19 The authors have felt it necessary to make the edge length 10 times the canister length to avoid biases from the boundary. Although I appreciate that this was no-doubt chosen because it was a round number, that was at least as large (but probably larger) than necessary, I would guess that within this domain one may find either the aircraft fuselage itself or air which had been modified by flow around the fuselage. It may well be that this is not the case, or that the flow distortion caused by the fuselage does not impact the particle paths, but the authors should show that this is the case or they should provide some limits on the potential effects of the fuselage.

It was computationally too expensive and therefore not feasible to include the entire aircraft fuselage in the simulations. However, the pylon where the probe is mounted is about 1.5 m away from the fuselage therefore the wing and the pylon/canister/instrument are the main contributors to flow distortions at the measurement location. Furthermore, the measured and simulated differences in pressure and air speed at the probe in comparison to free stream conditions show that the simulated values are well within the measured range (see also Figure 6 in the manuscript). Therefore, we think the exclusion of the fuselage does not introduce a significant bias.

They should also describe the effect of the wing itself, so that it is clear to the reader that both the wing and the instrument canister have an impact upon the flow. This will certainly feed in to decisions made by future studies into flow effects on microphysics measurements.

We performed new numerical simulations only considering the effect of the wing. The result is shown in Figure 2. The wing will contribute only up to 9% whereas the total effect is up to 24%.

[Figure]

*Figure 2. Same as Figure 8 in the manuscript, but only simulating the effect of the wing. The sampling efficiency is calculated as a function of modified Stokes number (see Eq. 4) for the selected numerical test cases of Tab. 3 in the manuscript. Each marker represents a run where we released $2 \times 10^5$ particles of a specific diameter (colors) and density (small marker size: water; large marker size: dust) calculated at the upstream of the probe in the computed flow field. Sampling efficiency is defined as the ratio between particles released and particles passing through the sampling area, renormalized by the corresponding areas.*

P8L14 It would be nice to highlight the test case on all plots, either by circling the data point or putting the line in bold or some other method.

The test case (u100_p900) is always plotted with blue color.

P8L21 replace probe and free stream with local and free stream

Done.

P8L28 it would be good to see the incompressible solution on the charts for comparison.

The incompressible solution will look like the u75_p1000 simulation where the effect of is still small. We added a statement in the text.

P9L1 At first reading I thought the 1% error in ps was causing a 23% error in U. It maybe worth rewriting this sentence to ensure other readers don't make the same mistake.

Done.

P9L2 How much do we really care about temperature? Later in the paper the authors refer to the fact that there is a compression of air and a bias proportional to density which is in turn proportional to temperature. However, the temperature increase is a maximum of 3%. The authors also mention that the temperature is measured "round the back" of the probe so may not be that relevant to the

sample volume anyway. The authors should simply consider if the bias caused by temperature increase is worth considering, given the other uncertainties, and if it is they should explain at this point why it is of interest.

The temperature bias has only a minor effect. We kept mentioning the temperature bias in the text but also made clear that for our case it only has a minor effect.

P9L11 "Well represent" is an ambiguous term. The points from the simulations on figure 6c do not show the upside-down U shape that the measurements show. The authors should state some measure of the actual discrepancy and why they feel that this is sufficient.

For small TAS (below ~90 m/s), the configuration of the aircraft may be different (typically these speeds are observed during take-off and landing where the flaps are used). Also, the effect from the ground might be not negligible at low TAS. This is why we are not surprised by the small deviation between simulations and observations at these TAS values.

P9L14 What does "installed at the back" mean? Give some better description of the location of the temperature sensor and if it is not close to the sample volume then describe the expected error due to the position and whether this is sufficient – see above comment re temperature in general.

Figure 3 (see below) shows the position of the CAPS temperature sensor. The sensor is located at the same position in case of the CAS instrument.

Figure 6 (left panel) in the manuscript shows a statistical analysis of differences in the ratios between temperature values measured with the Falcon CMET system and with the CAS instrument during SALTRACE.

[Figure]

CAPS temperature sensor

*Figure 3. Photograph of the CAPS mounted at the wing of the NASA research aircraft DC-8 during ATom-4. The position of the temperature sensor is marked by the white arrow. (Photograph: B. Weinzierl).*

P9L31 In what way is the stokes number modified and why?

According to Israel and Rosner (1982) the Stokes number is modified by introducing the correction factor psi which is a function of the Reynolds number distinguishing between the laminar and the fully turbulent case.

The text was modified accordingly.

P9l33 Equation (4) which velocity is used here for U? See later comments regarding stokes number.

To calculate equation 4 we use TAS. We added a new sentence to clarify the procedure.

P10 It is very impressive how well the data fit on the sigmoid curves on fig 8. This clearly forms an excellent correction factor. However, a few items may aid the reader in understanding the analysis that is occurring here.

Firstly, it would be good to indicate that the fit lines used are from equation (6).

The fit lines were obtained using a generic sigmoid function of the Stokes number using x0, k0 and k1 as free parameters. In the second step, we correlate these numbers (x0, k0, and k1) with flight conditions expressed as the alpha parameter.

We would also like to point out that there was typo in equation (6) where a plus was missing. We replaced it with the corrected equation.

Secondly, it should be noted that α is of the order 1 and contributes negligibly to the parameters x0 and k0, so should probably be removed.

The formulas given for x0, k0 and k1 were not correct. With the correct equations, the effect of alpha is stronger. Alpha is close to 1 in our case, but can have a larger effect in other cases. Therefore, we decided to keep the more general formula.

Thirdly, the authors should note that eˆ(-k0(log(stk)-x0) can be rearranged to b*stkˆa which is a much simpler form. Also if they remove the dependence of k0 and x0 upon α, then a and b simply become fitting constants. In fact b=10ˆ(x0*log(eˆk0)) and a=-log(eˆk0). Which makes b approximately -1.

We rearranged the formula.

Fourthly, it may be useful for the authors to rearrange the form slightly as this may help the points all converge onto 1 line.

We are not sure whether we understood this comment correctly. If the comment refers to Figure 8, we would like to clarify that the points in Figure 8 are not sitting on a line.

They should note the classic inlet efficiency equation from Belyaev & Levin doi:10.1016/0021-8502(74)90130-X, equation (5) Efficiency= 1+(u0/u-1) *sigmoid_function_of_stk_only Where u0 is

the far field velocity and u is the velocity at the inlet. This basically states that the deviation from unity efficiency is proportional to u0/u. This form may be useful to the authors, perhaps with u0/u replaced by α.

We considered the recommended explanation and the references in the text.

Fifthly, as described previously, the authors should consider whether it is really worth including temperature in this analysis as it is a relatively small effect and temperature may not be well measured by the probe.

See above.

P11L14 Do the authors actually mean mobility in the sense that it is used in aerosol science, e.g. for a scanning mobility particle sizer instrument? If not then change this word.

Yes, the term mobility is used in the same sense that it is typically used in aerosol science.

P11L26 The word positional error seems like an odd choice. Perhaps the authors mean distortion errors or something similar.

We improved the subsection title.

P11L27 did the authors mean 100 um? 10 um particles seem significantly affected by the flow.

We corrected the sentence now referring to particles with 30 μm diameter.

P11L31 rephrase. I think you want to say something like we adjust the calibration constants in the data logging software so that the pitot tube reports an air speed close to TAS rather than PAS. When this is reported to the instrument it causes the (insert info about lines imaged at the correct rate), but the PAS may still be recovered later by using the recorded pressures and the correct calibration constants.

We rephrased the sentence as recommended.

P12L14 The images shown in figure 12 are clearly distorted. However, it is not at all clear to me that this is a distortion of the droplets or just the images. In particular the images show a skew. This skew could be due to misalignment of the particle trajectory with the instrument axis, or perhaps it could be due to shear flow distorting the particles. This skew causes an apparent lengthening of the particles in the x direction – you will note that the farthest shadowed pixels on a droplet do not fall on the same line. It is not clear if any corrections for this have been made. Until a model can account for the skew distortion and the flattening, I think it is difficult to claim which mechanism is responsible. I think it is appropriate here to discuss how flattened the droplets are expected to be and to suggest it is a possible mechanism in the context of other possible mechanisms. But I don't feel it has been proven here that droplet flattening is the definite cause. Indeed the authors have the flow data to answer the question – are the particles travelling parallel to the probe axis?

The observation is correct. The particle trajectories especially for droplets with d>50μm can be considered parallel to the AOA. The difference between the AOA and the probe axis is in the

considered case small enough to be negligible. Indeed even considering a 1 degree angle between the AOA and the instrument axis this will result in a orthogonal component of TAS*sin(1°) which is less than 2% of TAS.

We would like to point out that only the larger droplets in Figure 12 are distorted, the smaller droplets (300nm and smaller) are not. This lead us to the conclusion that it is not the image what is distorted, but the distortion is caused by an aerodynamic force which preferentially acts on the larger droplets.

To make this point more clear, we included additional measurements from ATom-4 in Figure 12.

P13L5 It appears that airspeed errors of 10-20% would be perfectly sufficient to center the distribution of dy/dx over 1.0 for Atom-1. The a-life and Atom-2 data appear to show the opposite effect to that which would be expected from the droplet distortions described here. Please explain.

We included additional data from ATom-4 also in Figure 13.

We would like to point out that for ATom-1 the CAPS pitot tube was calibrated to measure the PAS, which for large particles leads to errors in the particle's/droplet's velocities of about 15% (depending on particle/droplet size). See Figure 11a in the manuscript for ATom-1. For A-LIFE, ATom-2 and ATom-4, we modified the calibration of the CAPS pitot tube to measure the TAS. The reason was to force the instrument to take the CIP images with the correct camera speed (large particle move with a velocity close to TAS).

If droplets start to deform by elongating in the y-dimension, they have a higher chance not to be fully recorded (see red contours in the new Figure 12 in the manuscript) and consequently they are removed from the analysis (Figure 13). This means that the deforming effect becomes less visible on larger particles.

P13L21 Be specific about what a small deviation is. Give the amount.

The results of our simulations are within two times the error bar of the experiments by Vargas (2012). We extended the text.

P14L23 There is no evidence for Taylor instabilities in Fig 13. Visually the scatter in the data looks to be entirely consistent with the error bars that the authors have put on the data. The authors would need to do some further analysis of the data scatter and if they find that the scatter is too large to be explained by the uncertainties, then and only then, should they invoke a mechanism to explain the extra scatter.

In the range between 200-500 µm, the dx/dy=1 line is outside the range of the error bar for most droplets. We think the data is sufficient to say that there is some scatter. We believe that the mentioned instabilities can be a possible reason for this scatter.

P14L28 Something here is not really matching up. The authors state that for a 200 um droplet they have We of 2.5 and it cannot be considered spherical, however close to that diameter on figure 13 the particles seem to generally have dy/dx within one error bar of unity. The author appears to be suggesting that observed broken up particles are caused by the mechanism here, or at least that some aircraft can go fast enough that droplets will be broken up so cannot be observed by the probes above a certain diameter.

Figure 12 shows deformed and broken particles/droplets (see red contours). According to Vargas (2012) this effect can be explained by aerodynamic forces.

But the points in Figure 15 show droplets going up to aspect ratios of 1.5 to 8. The only times I have seen such distorted particles has been when the refresh rate of the diode array has been clearly wrong. It may be that the authors have extrapolated the data for figure 15 past the break up point or perhaps there is something I am not able to piece together. But I can only suggest that section 3.2 needs a really thorough rewrite in order to ensure the arguments being made are consistent with the data and the model and to be clear what is a discussion and what is a conclusion.

We would like to point out that Figure 15 shows the results of simulations and not the analysis of droplets measured with the CIP instrument. In the grey-shaded area of Figure 15 there are also broken particles.

P14L29 I feel that the items discussed in section 3.2.2 are probably an over analysis of the data. Again the authors should be clear about what is a conclusion and what is a discussion and before they begin suggesting corrections for an effect they first need to be clear that they have shown the effect is real.

We would like to refer to Vargas (2012), who has shown the effect in a laboratory setting. Therefore, the shown effect is probably real. Furthermore, the simulations reproduce the same effect. Therefore, we think that a model-based testing of different formulas for the droplet volume calculation is useful for the reduction of errors of the droplet volume calculated from droplet images detected by OAP.

P15L25 should be clear that size is diameter rather than radius

Done. Everywhere, diameters are used.

P15L30 see comment above re corrections

See comment above.

P16L5 This set of steps would be much simplified to the point of being obvious if the advice above is taken regarding making the equations a bit clearer and specifying which air speed is being used for Stk.

We modified the manuscript and would like to point out that the velocity used for calculating Stk number in our case is the TAS.

P17L3 Again it is not clear to me that droplet deformation as a cause of the image distortion has been proven.

The images in Figure 12 show that smaller droplets appear circular while larger particles appear deformed. We extended Figure 12 to include droplet images from another campaign. They confirm the same finding. We extended the sentence.

P17L13 Stating this as a potential limit for the drop size that can be seen as a function of aircraft speed is I think very interesting, but my gut feeling is that observed droplet breakups are caused by impacts or near impacts with the probe.

In Vargas (2012), the effect has been observed in lab experiments. This lab experiment has been verified in our simulations.

P26Fig2 It would be nice to highlight the test case.

In all figures, the test case u100_p900 is marked in blue.

P27Fig3 The lines of best fit are linear – the word polynomial should be saved for higher order equations. Replace with Linear. All plots should at least have x and y axes that start at the same value. I would ideally like to see the same max value on the x and y axes too, but I appreciate that for Fig2d, this may not be appropriate.

Probably this comment refers to Figure 2? We updated Figure 2. Axes ranges for x and y axes start and end at the same values and "polynomial" was replaced with "linear".

P27Fig4 ensure that 0 is marked on the colour bar.

0 is now marked in the colour bar.

P28Fig5 It would be nice to see the test case highlighted

P29Fig6 It would be nice to see the test case highlighted

In all figures, the test case u100_p900 is marked in blue.

P36Fig13 The lines have no proper description of what they mean – are they averages over some size range? I think they are not useful anyway and should probably be removed unless the authors can argue a good reason for them. Also 2 different shades of grey should not be used. It is hard to tell them apart.

We modified the caption adding a description of the lines. The different campaigns are now plotted with 3 different colors and markers.

---

## Author Response (AR2)

Dear Claire Ryder,

thank you for your review and helpful comments. Please find below our responses.

In addition to the requested changes and several smaller changes we also performed the following changes:

- Equ. 4: Addition of missing units and a reference to formula for the correction factor ψ
- Equ. 5: Correction
- Equ. 6: Simplification of equation and update of fit coefficients

Please note, we are in contact with the data archive, but have not yet received the data doi (p.49, l. 811 in the manuscript). We expect to get the data doi in the next few days.

The manuscript with all changes marked since the previous version (submitted on 14 Jan 2020) is attached.

Kind regards,
Antonio Spanu on behalf of all coauthors

**Editor comments (blue) and author responses (black)**

1. Please ensure that all responses to the referees' comments are actually included in the manuscript itself. The referees' comments serve to improve the manuscript and also answer questions which other readers may have when reading the manuscript.

As suggested we have double-checked whether we have considered the responses to the referee comments in the new manuscript version.

- Added on page 7 some text about flow differences between different mounting positions and the flow around DC-8 vs Falcon
- Added a sentence on page 6 about conservation of total pressure at the two pitot tubes (reviewer 2).
- Added a sentence on page 10 about flaps effect at low TAS (reviewer 2).

Specifically, the following two points should be dealt with:
a. The response to both referees regarding the effects of the fuselage do not appear to have been accounted for in the manuscript. This should be actioned by the authors.

We now mention the distance of CAS from the fuselage (1.5m). We have added a sentence on page 10 about the estimated uncertainty due to the omission of fuselage and the simplified geometry.

b. Likewise, the angle of attack (AOA) comments, although dealt with acceptably in the response to reviewers, does not appear to have been justifiably transferred to the manuscript. This should be done.

We extended the justification for the angle of attack in the manuscript by adding into the appendix as a new Figures A2 and A3 the Figures 1 and 3 of our response to reviewer 1 which show the distribution of the angle of attack as function of TAS and that it has only a minor impact on the measurements.

c. Instrument geometry – impact of hemispherical domes vs actual probe design – please ensure your response to reviewers is reflected in the manuscript, explaining why no need for specific instrument geometry is required in this study.

We have added a sentence on page 10 about the uncertainty due to the omission of fuselage and the simplified geometry indicating the validity of the instrument geometry in the model.

2. Figure 12 has been changed to show observations from a different campaign. This is different to it being 'extended' or 'including additional measurements' as stated in the authors' response. Perhaps this is an oversight by the authors. The swap of data/campaign must be properly stated and justified, or both the original and new images should be shown in Figure 12, with the text discussing both datasets.

The main message of this figure is that very large particles are squeezed. This message was not clearly visible in the original figure of the discussion paper (with A-LIFE recordings) because the largest particles were around 500-600µm in diameter. During the time of paper submission, the Atom-4 data were not yet available. But, as we found out between our replies to the reviewers and submission of the revised manuscript, the Atom-4 images show the effect much clearer because water droplets with diameters of 1000µm and larger were recorded. We missed to explain that change during the submission of the revised manuscript. To make the message of Fig. 12 as clear as possible we now suggest to show only Atom-4 images since there is no added value in showing the A-LIFE images.

3. Regarding the CAPS temperature sensor 'installed at the back' – add further clarification to the manuscript itself.

We have added a photo in the appendix showing the CAPS temperature sensor. It illustrates the mounting position of the sensor on the instrument. We also added a reference to this figure in the text.

4. Respond to Referee 2's comment about p14L28 from the original ACPD manuscript in more detail, fully explaining the apparent inconsistency. (regarding the 200 micron droplets).

Our response was incomplete. Regarding the 200µm droplets in the discussion paper (at P14L28) we intended to express that in this size range so-called Tylor instabilities can occur which result in oscillations on the droplet surface which destroy the spherical shape of the droplet. As dy/dx fluctuates around a mean value (which is typically around unity if particles are not too large as in case of 200µm), these oscillations would result in some scattering of recorded dy/dx data (in addition to the scattering that originates from the limited spatial resolution of the recorded images). We reformulated part of Sect. 3.2.1 (page 16) to make this aspect more clear.

5. Figure 14 – the top panel (greyscale and droplets) appears distorted.

The upper part is the experimental data from the Vargas experiment, the lower part is the result of our simulations. To make things more clear, we modified the caption of Figure 14:

[revised manuscript text omitted]